

# Characterization and source apportionment of organic aerosol at 260 m on a meteorological tower in Beijing, China

Wei Zhou[1,2], Qingqing Wang[1], Xiujuan Zhao[3], Weiqi Xu[1,2], Chen Chen[1], Wei Du[1,2], Jian Zhao[1,2], Francesco Canonaco[4], André S. H. Prévôt[4], Pingqing Fu[1], Zifa Wang[1], Douglas R. Worsnop[5] and Yele Sun[1,2,6]

[1]State Key Laboratory of Atmospheric Boundary Layer Physics and Atmospheric Chemistry, Institute of Atmospheric Physics, Chinese Academy of Sciences, Beijing 100029, China

[2]University of Chinese Academy of Sciences, Beijing 100049, China

[3]Institute of Urban Meteorology, China Meteorological Administration, Beijing 100089, China

[4]Laboratory of Atmospheric Chemistry, Paul Scherrer Institute, Villigen PSI 5232, Switzerland

[5]Aerodyne Research, Inc., Billerica, MA, USA

[6]Center for Excellence in Regional Atmospheric Environment, Institute of Urban Environment, Chinese Academy of Sciences, Xiamen 361021, China

*Correspondence to*: Yele Sun (sunyele@mail.iap.ac.cn)

**Abstract.** Despite extensive efforts into characterization of submicron aerosols at ground level in the megacity of Beijing, our understanding of aerosol sources and processes at high altitudes remains less understood. Here we conducted a three-month real-time measurement of non-refractory submicron aerosol (NR-PM$_1$) species at a height of 260 m from 10 October 2014 to 18 January 2015 using an aerosol chemical speciation monitor. Our results showed a significant change in aerosol composition from non-heating period (NHP) to heating season (HP). Organics and chloride showed clear increases during HP due to coal combustion emissions, while nitrate showed substantial decreases from 28% to 15–18%. We also found that NR-PM$_1$ species in heating season can have average mass differences of 30–44% under similar emission sources yet different meteorological conditions. Multi-linear engine 2 (ME-2) using three primary organic aerosol (OA) factors, i.e., fossil fuel-related OA (FFOA) dominantly from coal combustion emissions, cooking OA (COA), biomass burning OA (BBOA) resolved from ground high-resolution aerosol mass spectrometer measurements as constrains was performed to OA mass spectra of ACSM. Two types of secondary OA (SOA) that were well correlated with nitrate and chloride/CO, respectively, were identified. SOA played a dominant role in OA during all periods at 260 m although the contributions were decreased from 72% during NHP to 58–64% during HP. The SOA composition also changed significantly from NHP to HP. While the contribution of oxygenated OA (OOA) was decreased from 56–63% to 32–40%, less oxidized OOA (LO-OOA) showed a large increase from 9–16% to 24–26%. COA contributed a considerable fraction of OA at high altitude, and the contribution was relatively similar across different periods (10–13%). In contrast, FFOA showed a large increase during HP due to the influences of coal combustion emissions. We also observed very different OA composition between ground level



and 260 m. Particularly, the contributions of COA and BBOA at ground site were nearly twice those at 260 m, while SOA at 260 m was ~15–34% higher than that at ground level. Bivariate polar plots and back trajectory analysis further illustrated the different source regions of OA factors in different seasons.

## 5  Introduction

Atmospheric aerosol particles reduce visibility by scattering and absorbing solar radiation, and also exert detrimental effects on human health (Dockery et al., 1993; IPCC, 2013). Beijing, the capital of China, has been suffering severe haze pollution since the last decade, especially in autumn and winter (Huang et al., 2014; Guo et al., 2014). For example, Beijing released the first air pollution red alert due to a severe haze episode in December 2015 when the daily $PM_{2.5}$ concentration exceeded

$300 \ \mu g \ m^{-3}$ for more than 3 days. Therefore, the sources and formation mechanisms of haze episodes have been extensively investigated for mitigating air pollution in Beijing in recent years (Huang et al., 2014; Sun et al., 2016a; Zhao et al., 2013b; Yang et al., 2015; Wu et al., 2017). The results highlight that stagnant meteorological conditions with shallow planetary boundary layer (PBL) and low surface wind, traffic and coal combustion emissions, regional transport and secondary productions of SNA (sulfate, nitrate, and ammonium) are the major factors leading to the formation of severe haze episodes.

However, our knowledge of the characteristics of aerosol particles is far from complete, mainly due to the fact that most previous studies were conducted at ground sites and were subject to strong influences from local emissions. The measurements and analysis at high altitudes in urban sites are still very limited, particularly long-term analysis by covering different seasons.

Organic aerosol (OA) accounts for a major mass fraction (20–90%) of submicron aerosol (Zhang et al., 2007), and

comprises primary OA (POA) from direction emissions and secondary OA (SOA) from oxidation of volatile organic compounds (VOCs) (Jimenez et al., 2009). The deployments of Aerodyne Aerosol Mass Spectrometer (AMS)/Aerosol Chemical Speciation Monitor (ACSM) for real-time measurements of aerosol particle composition followed by subsequent positive matrix factorization (PMF) analysis (Paatero and Tapper, 1994; Ulbrich et al., 2009) have greatly improved our understanding on the sources of OA in China, and also highlight the important role of OA in the rapid formation of severe

haze (Huang et al., 2014; Sun et al., 2014; Zhang et al., 2014). The sources and formation of OA are seasonally different. For example, SOA is more significant than POA in summer due to stronger photochemical processing, while POA often plays a more important role than SOA in winter due to more primary emissions, e.g., coal combustion, and weaker photochemistry (Sun et al., 2015b; Li et al., 2017; Ma et al., 2017). A recent study by Xu et al. (2017) further found that photochemical processing tends to form less oxidized SOA (LO-OOA), while aqueous-phase processing can form more oxidized SOA

(MO-OOA).

Despite extensive OA studies in recent years, our understanding of the sources and processes of OA at a high altitude in megacities is still limited. Sun et al. (2015a) conducted the first real-time measurements of aerosol particle composition at 260 m on the Beijing 325 m meteorological tower (BMT) using an ACSM. A high-resolution time-of-flight AMS (HR-AMS)





was deployed in parallel at ground level during the same period. The results showed substantially different aerosol composition between 260 m and ground level, particularly, OA showed higher contribution to non-refractory $PM_1$ (NR-$PM_1$) at ground level (65%) than 260 m (54%). Source apportionment of OA further illustrated the similar temporal variations of SOA at the two heights, while those of POA were dramatically different. Also, the contribution of SOA to OA at higher

altitude was higher than that at ground site (49% vs. 38%). Chen et al. (2015) further analyzed OA at 260 m before and during Asia-Pacific Economic Cooperation (APEC) summit when strict emission controls were implemented in Beijing and surrounding regions. The results showed similar reductions in POA and SOA at 260 m during APEC as a response of emission controls. However, POA remained at small changes at ground level although SOA showed similar reductions as that at 260 m. Such differences highlight the different impacts of local source emissions and regional transport on POA and

SOA at different altitudes in the city. Note that PMF analysis of OA at 260 m in previous studies was all limited to a 2-factor solution, i.e., POA and SOA. Although the mass spectrum of POA shows a mix of different primary emissions, e.g., traffic, cooking and biomass burning, the solutions with more factors however showed unrealistic split of OA factors (Sun et al., 2015a; Chen et al., 2015). As a consequence, the variations and evolution of different POA and SOA factors are poorly understood at a high altitude in Beijing. For example, Zhao et al. (2017) found that the two different SOA factors, i.e.,

LO-OOA and MO-OOA showed different responses to emission controls during the 2015 China Victory Day parade at ground site. While MO-OOA showed a large decrease during the control period, LO-OOA was comparable during and after the control period. However, the composition and variations of different SOA factors at 260 m were never characterized. This also greatly limits our understanding of the sources and processes of OA at high altitudes in the planetary boundary layer in Beijing.

In this study, an ACSM was deployed at 260 m on the BMT for three months to measure NR-$PM_1$ aerosol species in real-time. This study encompasses three periods with different emission scenarios, i.e., non-heating period (NHP), APEC with strict emission controls in Beijing and surrounding regions, and heating period (HP) with significant influences from coal combustion emissions. The characteristics and sources of NR-$PM_1$ and OA before and during APEC were characterized in our previous study (Chen et al., 2015), here we mainly focus on characterization of submicron aerosols during the heating

season, and also the comparisons with those during NHP and APEC. Most importantly, we present the first source apportionment analysis of OA at 260 m by using the multi-linear engine (ME-2) with the constrained POA factors identified at ground site. The mass concentrations, composition, and diurnal cycles of POA and SOA factors are characterized, and the changes in POA and SOA composition from NHP to APEC and HP are elucidated. Also, the comparisons of sources and composition of POA and SOA between ground level and 260 m are presented.

## 2    Experimental methods

### 2.1    Sampling site and measurements





The NR-PM$_1$ species, including organics (Org), sulfate (SO$_4$), nitrate (NO$_3$), ammonium (NH$_4$), and chloride (Chl), and gaseous species of SO$_2$ and CO, were measured at the height of 260 m on the BMT (39°58′28″N, 116°22′16″E, ASL: 49 m) using an ACSM and gas analyzers (Thermo Scientific). A more detailed description of the sampling site and the operations of the ACSM can be found in Sun et al. (2015a) and Chen et al. (2015). Simultaneously, a HR-AMS was deployed at the ground level to measure the size-resolved NR-PM$_1$ composition (Xu et al., 2015). In addition, the meteorological variables including wind direction (WD), wind speed (WS), relative humidity (RH), and temperature (*T*) at 8 m and 280 m were also measured on the BMT.

### 2.2   ACSM data analysis

The mass concentrations and chemical compositions of NR-PM$_1$ were analyzed with the ACSM standard data software (V1.5.3.0). Consistent with our previous study in Beijing in 2014 (Chen et al., 2015), we used the same parameters for the entire campaign. For example, a constant collection efficiency (CE) of 0.5 was applied to compensate for the incomplete detection of aerosol particles which is primarily caused by particle bounce effects at the vaporizer (Matthew et al., 2008). The reason is that the mass fraction of ammonium nitrate, RH and particle acidity were found to play minor influences on CE in this study (Middlebrook et al., 2012). Note the ACSM NR-PM$_1$ species were further corrected using the corresponding scaling factors (0.61–1.24) that were determined from the two week inter-comparisons between ACSM and HR-AMS.

### 2.3   Positive matrix factorization

Positive matrix factorization (PMF) was first performed to the unit mass resolution spectra of OA (*m/z* 12–350) at ground site that were measured with HR-AMS during the same period as that of ACSM. Five factors, i.e., a fossil fuel-related OA (FFOA) from traffic and coal combustion emissions, a cooking-related OA (COA), a biomass burning OA (BBOA), an oxygenated OA (OOA), and a less oxidized OOA (LO-OOA) were determined. Extending the PMF solution to 6 and 7 factors still cannot separate traffic hydrocarbon-like OA (HOA) from coal combustion OA (CCOA). The mass spectra of five OA factors are presented in Fig. S1. Each factor was characterized by the typical prominent peaks indicative of different sources and properties (Sun et al., 2016b), for example, high *m/z* 60 and 73 peaks in BBOA, prominent PAHs fragments in FFOA, high *m/z* 55/57 in COA, and pronounced *m/z* 44 in SOA (LO-OOA and OOA). The mass spectral profiles of the three POA factors, i.e., FFOA, COA, and BBOA were then used as the constrains of multi-linear engine 2 (ME-2) analysis at 260 m. Compared with the PMF2 algorithm, ME-2 reduces rotational ambiguity by adding the known source information (e.g., factor profiles or factor time series) into the model (Canonaco et al., 2013; Paatero and Tapper, 1994). Considering the POA factors at 260 m might not be completely the same as those at ground site, the so-called a-value approach with a-value ranging from 0 to 0.5 for all three POA factors were performed to OA. Following the guidelines of the a-value sensitivity test presented by Crippa et al. (2014), we evaluated the mass spectral profiles, diurnal patterns, time series of OA factors, and also compared with external tracers. As shown in Fig. S3, the time series of FFOA and COA were fairly robust across





different a-values, and the differences were generally less than 10%. Although the COA spectrum was also robust, the FFOA spectrum showed a clear decrease of the ratio of $m/z$ 41/43 to $m/z$ 55/57 as a-value increases, and became more different from that of burning smoky coal (Lin et al., 2017). Compared with FFOA and COA, BBOA showed more variability in both spectrum and time series. As a-value increased from 0 to 0.5, the average BBOA concentration was increased by nearly a

5 factor of 2, and the mass spectrum showed a significant decrease of $m/z$ 60. To better compare with the PMF results at ground site and also allow for some degrees of freedom for model runs, the five-factor solution with a-value of 0 for FFOA and BBOA and 0.2 for COA was selected in this study. We also performed PMF analysis on ACSM OA spectra, and found that the BBOA factor cannot be resolved although biomass burning is a common source in winter. Figure S4 shows a diurnal comparison of the solutions between PMF and ME-2. It is clear that the unconstrained PMF reports twice higher FFOA and

10 COA than ME-2, while OOA is 30% lower (Fig. S2). While the rotational ambiguity is one of the reasons, another explanation is that FFOA and COA are mixed with BBOA which cannot be resolved by PMF. More detailed ME-2 diagnostics are presented in Figs. S2–S4.

## 3    Results and discussion

### 3.1    Characteristics of NR-PM$_1$ species at 260 m

#### 3.1.1  Mass concentrations and chemical composition

Figure 1 shows the time series of meteorological parameters and NR-PM$_1$ species for the entire study, which can be classified into three different periods, i.e., NHP, APEC and HP according to the changes in source emissions. The HP was further separated into two different periods according to the variations in $T$ and RH. As shown in Fig. 1, the first HP period (HP1) from 13 November to 31 November showed clearly higher RH and $T$ compared with the second period (HP2) from 1

December to 18 January. The NR-PM$_1$ mass concentration varied significantly throughout the entire study with hourly average concentration ranging from 0.7 to 284.8 μg m$^{-3}$. Such dramatic variations primarily driven by meteorological changes at ground sites have also been observed many times in autumn and winter in Beijing (Sun et al., 2013b; Guo et al., 2014; Zhang et al., 2016b). For example, the changes in NR-PM$_1$ were characterized by routine cycles of clean periods and polluted episodes during HP2. Comparatively, polluted episodes that lasted approximately four days and then followed by a

short period of clean days were more frequent during NHP, mainly due to the fewer occurrences of northwesterly winds (Fig. S5). Consistently, periods with high NR-PM$_1$ loading (> 80 μg m$^{-3}$) accounted for more time during NHP (37%) than HP2 (25%). Indeed, the average NR-PM$_1$ concentration during HP2 (48.6 μg m$^{-3}$) was even lower than that during NHP (64.9 μg m$^{-3}$), illustrating that the PM pollution in autumn 2014 was more severe than that in winter. The large reduction of NR-PM$_1$ by 61% during APEC compared with that during NHP due to regional emission controls and favorable meteorological

conditions has been reported in Chen et al. (2015) and Sun et al. (2016c). This is also consistent with the fact that the periods with NR-PM$_1$ less than 60 μg m$^{-3}$ accounted for 93% of the time during APEC, which is much higher than that before APEC



(51%).

The two heating periods showed very different PM levels with the average NR-PM$_1$ concentrations being 74.5 and 48.6 µg m$^{-3}$, respectively. As shown in Fig. 2, the frequency of NR-PM$_1$ mass during HP1 was significantly different from that during HP2. Clean periods with NR-PM$_1$ < 20 µg m$^{-3}$ accounted for nearly half of the total time during HP2, while it was
much less during HP1 (34%). Considering that emission sources could not have significant changes during the heating season, such differences were mainly caused by the different meteorological conditions, for example more frequent north-westerly winds during HP2 than HP1 (Fig. S5). In addition, the different RH and $T$ also indicate the different chemical processing, e.g., photochemical and aqueous-phase processing between HP1 and HP2. Note that the average NR-PM$_1$ during HP1 was only 15% higher than that during NHP, and that of HP2 was even 25% lower, which appears to contradict with our
previous conclusion that HP showed approximately 50% higher PM loading than NHP (Wang et al., 2015). One major reason is due to more frequent clean periods during HP in 2014 than 2012 (Wang et al., 2015).

Organics comprised the major fraction of NR-PM$_1$ during all periods, on average accounting for 46–54%. The dominance of organics in NR-PM$_1$ was consistent with previous studies at ground sites in Beijing (Sun et al., 2013a; Huang et al., 2014; Zhang et al., 2016a). Note that the contribution of organics showed an increased during HP as a result of
enhanced emissions from coal combustion (Elser et al., 2016; Wang et al., 2015; Zhang et al., 2016b). Consistently, the combustion-related chloride was increased by a factor of ~2 from 4% during NHP to 7–8% during HP, and the mixing ratio of SO$_2$ was increased from 8.0 to 12.6–13.9 ppb. Our results highlight the great impact of coal combustion on aerosol chemistry at a high altitude in the city. The two secondary inorganic species of sulfate and nitrate showed different variations between NHP and HP. The sulfate contribution during HP (14–15%) remained at similar levels as that during NHP (14%),
and the concentrations were also relatively close (6.9–11.2 µg m$^{-3}$ vs. 8.8 µg m$^{-3}$), while the nitrate contributions and mass concentrations decreased substantially from NHP (28% and 17.9 µg m$^{-3}$) to HP (15–18% and 7.4–13.3 µg m$^{-3}$).

Such differences demonstrate the different formation mechanisms of nitrate and sulfate in different seasons. Nitrate appears to be mainly driven by the photochemical production in wintertime while sulfate is mainly from aqueous-phase/heterogeneous reactions (Sun et al., 2013a). The recent study further highlight that the oxidation of SO$_2$ by
NO$_2$ in aerosol water in the major mechanism (Cheng et al., 2016). As shown in Fig. 3, the SO$_4$/NO$_3$ ratios increased similarly along with RH during all periods, indicating that aerosol liquid water exerted a larger impact on the formation of sulfate than nitrate. The large increases in sulfate and SOR (sulfur oxidation ratio) during periods with high RH, for example, SOR was increased from 0.09 (RH = ~45%) to 0.48 (RH = ~85%) during HP1, further supporting the aqueous-phase production of sulfate (Ohta and Okita, 1990). The SO$_4$/NO$_3$ ratios were ubiquitously lower than 1 during NHP and APEC,
suggesting a more important role of nitrate in PM pollution in autumn. Comparatively, sulfate gradually exceeded nitrate as RH was increased to > 60% during HP, and became the dominant secondary inorganic species. Also note that the SO$_4$/NO$_3$ ratios showed large variations (~0.2–10) during periods with low sulfate mass loading (< 3 µg m$^{-3}$), indicating more source variability during clean periods.



### 3.1.2 Diurnal variations of NR-PM$_1$ species

The diurnal variations of meteorological variables, NR-PM$_1$ species and SO$_2$ are presented in Fig. 4. A detailed comparison of the diurnal patterns between NHP and APEC was presented in Chen et al. (2015). As shown in Fig. 4, the diurnal patterns of meteorological parameters (RH and WS) were relatively similar between NHP and HP1 except lower $T$ during HP1. However, the diurnal patterns of organics, sulfate, and chloride showed ubiquitously higher concentrations during HP1 than NHP throughout the day, supporting the significant influences of coal combustion emissions on these three species (Wang et al., 2015). Two clear organic peaks occurring at lunch and dinner time were observed during both periods, highlighting the influences of cooking emissions at high altitude in Beijing. Note that the differences of organics at nighttime (~30–50%) were larger than those (< 30%) during daytime, consistent with the enhanced coal combustion emissions at nighttime. We also noticed that all NR-PM$_1$ species during HP1 showed similar diurnal behaviors that were characterized by first gradual decreases from mid-night to early morning (~8:00 am), and then increases during the rest of the day. We found that such diurnal patterns were strongly associated with the daily changes in wind direction. For example, the winds were dominantly from the north at nighttime, while they switched to the south after 12:00. The increases between 8:00–12:00 can be explained by the corresponding decreases in wind speed (Fig. 4).

The diurnal patterns of NR-PM$_1$ species during HP2 were similar to those during HP1, yet the mass concentrations were much lower. For example, nitrate showed the largest decrease by 34–55% followed by sulfate (27–48%) and organics (13–50%) during HP2. Considering the similar emission sources between HP1 and HP2, such decreases can be explained by more frequent north-westerly winds that were associated with higher wind speed and lower RH during HP2 (Fig. 4). This is further supported by the fact that the polluted episodes showed much smaller differences between HP1 and HP2 (e.g., ~20% for NR-PM$_1$, discussed in Sec.3.3). In addition, the reduced photochemical processing as indicated by the decrease in nitrate during HP2 also played a role. We also note that SO$_2$ at nighttime during HP2 was even higher than that during HP1, suggesting less transformation of SO$_2$ into sulfate due to drier conditions (RH = ~30%). Our results demonstrate that meteorological condition is a critical factor affecting the PM levels during heating season in addition to coal combustion emissions. For example, the favorable meteorological condition can cause an average decrease of NR-PM$_1$ by 35%, varying from 26 to 44% for different aerosol species (Table 1). Therefore, it is critically important to exclude the meteorological effects for an accurate evaluation of the impacts of source emission changes on air quality.

## 3.2 OA composition, sources and variations

### 3.2.1 Fossil fuel-related organic aerosol (FFOA)

The FFOA spectrum shows prominent hydrocarbon ion series C$_n$H$_{2n+1}^+$ (*m/z* 29, 43, 57) and C$_n$H$_{2n-1}^+$ (*m/z* 27, 41, 55), yet the spectral pattern resembles much more to that of burning of smoky coal (Lin et al., 2017) than the standard traffic-related HOA (R$^2$ = 0.79) (Ng et al., 2011). These results suggest that FFOA at 260 m is likely dominantly from coal combustion emissions rather than traffic emissions, consistent with the better correlations between FFOA and chloride and SO$_2$ during





HP than NHP (Fig. 6). FFOA was also tightly correlated with CO, a tracer for combustion emissions, during HP ($R^2$ = 0.62–0.68). As shown in Fig. 5a, the temporal variation of FFOA showed a dramatic increase after the heating season starts on 15 November. The average concentration of FFOA during HP1 and HP2 was 4.7 and 5.2 µg m$^{-3}$, respectively, which is more than 3 times higher than those during NHP and APEC (1.2–1.4 µg m$^{-3}$), indicating the largely enhanced coal combustion emissions during the heating season. Correspondingly, the FFOA contributions in OA also showed large increases from 5% during NHP to 13–21% during HP. FFOA showed a pronounced diurnal cycle during HP1 and HP2 with nearly twice higher concentration at nighttime than daytime. However, the diurnal pattern was less significant during NHP, indicating the dominant source of regional transport for FFOA at 260 m. The bivariate polar plot of FFOA (Fig. 9a) further indicates that high concentration of FFOA during NHP was dominantly from the regional transport in the southwest. Consistently, the FFOA concentration at 260 m was much higher than that at ground level throughout the day (average: 0.4 µg m$^{-3}$) during NHP when coal combustion emissions were not important inside the city. The diurnal patterns of FFOA were substantially different between ground level and 260 m during HP. Particularly, the diurnal cycles at ground level showed much larger day and night differences. As shown in Figs. 7c and 7d, the FFOA concentrations at ground level dropped rapidly at ~3:00–4:00 and reached the minimum at ~14:00. One explanation is the largely enhanced coal combustion emissions at nighttime in local areas. In comparison, the diurnal cycles of FFOA at 260 m were much smoother, and the concentrations in the daytime were even higher than those at ground level We also noticed slightly higher FFOA concentration during HP2 (5.2 µg m$^{-3}$) than HP1 (4.7 µg m$^{-3}$). The bivariate polar plots in Fig. 9a showed a high FFOA concentration (> 12 µg m$^{-3}$) region in the southwest during HP2, while it was negligible during HP1 but primarily from the local area. Such a difference in source regions explained the higher FFOA concentration during HP2 than HP1.

### 3.2.2 Cooking organic aerosol (COA)

COA that contributes a large fraction of OA at ground level in megacities has been widely characterized (Crippa et al., 2013; Mohr et al., 2012; Allan et al., 2010). However, the characteristics of COA at high altitudes in megacities were poorly characterized. Our results showed similar diurnal cycles of COA at 260 m to those observed in Beijing and other urban sites (Sun et al., 2012; Xu et al., 2014; Huang et al., 2010), which were characterized by two pronounced peaks at meal time. The noon COA peak at 260 m was relatively comparable to that at ground level, mainly due to the vertical mixing associated with the rising boundary layer during daytime. In fact, the COA concentrations were almost the same between ground level and 260 m in the late afternoon (~16:00). In contrast, the night COA peak at 260 m was significantly lower than that at ground level. One explanation is the shallow boundary layer and frequent temperature inversions at night suppressed the vertical mixing of COA to a high altitude. Figure 9b shows that high concentration of COA was mainly located in a small region near the sampling site during all periods, highlighting the dominant local sources of COA. Indeed, the COA mass concentrations were relatively comparable (2.7–4.8 µg m$^{-3}$) across different periods except APEC (1.2 µg m$^{-3}$) with strict emission controls (Fig. 5). This is consistent with the fact that cooking emissions are relatively stable during all seasons. The average





contributions of COA were also close during four periods, ranging from 10–13%, yet ubiquitously lower than those (18–38%) observed at ground site (Xu et al., 2015). Previous studies have shown that the impact of COA is spatially limited as the concentration decreased rapidly outside of the city (Ots et al., 2016). Our results illustrated the influences of cooking emissions on OA at high altitudes in megacities, and the impacts are expected to be vertically limited as the concentration

decreased with the increasing height.

### 3.2.3 Biomass burning OA (BBOA)

The average concentrations of BBOA were comparable between NHP (3.0 μg m$^{-3}$) and HP (2.7–3.2 μg m$^{-3}$) suggesting that biomass burning was an important source of OA at 260 m in both autumn and winter. Indeed, BBOA accounted for a considerable fraction, 11% and 9–11% during NHP and HP, respectively, of total OA. The BBOA concentration at 260 m

was decreased by ~70% during APEC, which is much higher than that (16%) at ground site (Xu et al., 2015). One explanation is that BBOA during APEC was more from regional transport at 260 m (Fig. 9c) while there were still considerable local biomass burning emissions near the sampling site. Note that BBOA was better correlated with secondary inorganic species during NHP and HP (R$^2$ = 0.54–0.80) than chloride (Fig. 6), suggesting that BBOA at 260 m was likely relatively well mixed with secondary species over a regional scale. This is also consistent with their similar diurnal patterns

(Fig. 4). Although the contributions of BBOA to OA were comparable (7–11%) during the four periods, the sources can be very different. For example, BBOA was mainly from the southwest and southeast of Beijing during NHP, while the concentrations were low in the nearby regions (Fig. 9c). These results suggest that regional transport was the major source of BBOA at 260 m during NHP. Comparatively, BBOA during APEC and HP2 were from both local emissions and the transport from the southwest while it was primarily from local source emissions during HP1. The diurnal variations of

BBOA at ground level also showed much higher concentrations at nighttime than daytime during all periods, supporting the strong local biomass burning sources. Indeed, BBOA at ground level accounted for 20–28% of OA during HP, which is more than twice that of at 260 m (9–11%).

### 3.2.4 Secondary organic aerosol (OOA and LO-OOA)

The mass spectra of two SOA (LO-OOA and OOA) factors were both characterized by the prominent peak of $m/z$ 44 (mainly

CO$_2^+$) (Aiken et al., 2009). OOA showed much higher $f44$ than LO-OOA (0.28 and 0.09, respectively), indicating that OOA was more oxidized than LO-OOA (Aiken et al., 2008). Note that LO-OOA was highly correlated with large $m/z$'s (Fig. 6) which appears different from previous findings that POA typically correlates better with high $m/z$'s (Sun et al., 2016b). We noticed that LO-OOA showed much better correlations with external trace species during HP than NHP and APEC. Particularly, LO-OOA was well correlated with chloride (R$^2$ = 0.71–0.81) and CO (R$^2$ = 0.79–0.84) that were dominantly

from coal combustion emissions in heating season. In addition, the mass spectrum of LO-OOA presented a much higher $f43/$ $f44$ ($f43$, fraction of $m/z$ 43) ratio than previous studies (Wang et al., 2016). All these facts suggest that LO-OOA here was




very likely a combustion-related SOA. Consistently, the LO-OOA mass concentration showed a large enhancement from 4.6 and 1.1 µg m$^{-3}$ during NHP and APEC to 8.5 and 6.6 µg m$^{-3}$ during HP1 and HP2, respectively, supporting more fresh SOA production due to enhanced combustion emissions. We also note that the LO-OOA at 260 m was nearly twice that of ground site during NHP and HP1, while they were comparable during HP2. These results might indicate that the polluted conditions

with high RH are subject to form more LO-OOA at high altitudes.

OOA dominated OA during both NHP and APEC, accounting for 56% and 63%, respectively. In contrast, the contribution of OOA was largely reduced during HP, accounting for 32–40% of OA. This result indicates a large reduction in production of more oxidized SOA in heating season. As shown in Fig. 6, OOA was highly correlated with nitrate during all four different periods ($R^2 = 0.70$–$0.90$), indicating that OOA is likely dominantly from photochemical production as that of

nitrate (Sun et al., 2013b). Figure 7 shows that OOA increased significantly in the afternoon due to photochemical processing while that of LO-OOA was less significant. We noticed that OOA was correlated with sulfate during NHP and APEC ($R^2 = 0.73$ and $0.67$), and also periods with high RH during HP1 and HP2 (Fig. S6). These results together indicate that OOA likely contains two different types of SOA associated with photochemical and aqueous-phase processing, respectively, yet it is difficult to separate an aqueous-phase SOA factor as that of HR-AMS due to the limited sensitivity of

ACSM and limited specificity of the mass spectra (Sun et al., 2016b).

Overall, OA composition at 260 m varied substantially during the four periods. SOA dominated OA during all periods although the contributions were decreased from 72% during NHP to 58–64% during HP. The SOA composition also changed significantly, particularly LO-OOA showed large increases from 9–16% during NHP to 24–26% during HP. Correspondingly, the OOA contribution was decreased from 56–63% to 32–40%. Compared to the ground site, OA showed 15–34% higher

SOA contribution at 260 m, highlighting the importance of SOA at high altitudes in urban areas. In contrast, POA (= FFOA + COA + BBOA) played a more important role in PM pollution at ground during HP by accounting for 57–65%. Particularly, the contributions of COA and BBOA at ground site were nearly twice those of at 260 m. The large differences in OA composition between ground level and 260 m have significant implications that measurements at high altitudes are of great importance for validating and improving the model simulations of POA and SOA in the future studies.

**3.3   Comparisons between clean and polluted episodes**

To evaluate the different roles of aerosol species in PM pollution, we further compared aerosol compositions between clean periods and polluted episodes (Fig. 10). The average NR-PM$_1$ mass concentrations varied from 4.0 to 10.4 µg m$^{-3}$ during clean episodes with much higher concentrations during HP than NHP. Analysis of the compositional differences showed that FFOA and LO-OOA are two species with the largest enhancements during HP, supporting the influences of coal combustion

emissions (Fig. 8). Although aerosol bulk composition was relatively similar among different clean episodes, OA composition showed significant changes from NHP to HP, which are characterized by large increases in FFOA from 7–8% to 14–18%, and LO-OOA from 10–11% to 17–19%. OOA showed corresponding decreases from 60–64% to 44–50%.



Aerosol composition was quite different between clean periods and polluted episodes. For example, organics accounted for a higher fraction (55–60%) in NR-PM$_1$ during clean periods than that (46–54%) in polluted episodes. Higher contribution of organics during clean periods was also previously observed at the same ground site (Sun et al., 2013b). On the contrary, the nitrate contribution showed a large increase in polluted days during NHP and APEC, e.g., 29% vs. 12%, further indicating

the predominant role of nitrate in severe haze pollutions particularly in periods free of intense coal combustion. Also note that the differences in nitrate contributions during HP was small between polluted and clean episodes, for example, 16–18% in polluted episodes and 12–15% in clean episodes. These results suggest that the polluted conditions during HP do not facilitate the nitrate formation substantially. Compared with nitrate, the sulfate mass fractions didn't change much during NHP and HP with slightly higher contributions in clean episodes, indicating that sulfate was similarly important during both

clean periods and polluted episodes. We also noticed nearly twice higher chloride contribution in polluted days compared with clean periods during HP, confirming the increasing role of coal combustion emissions in severe winter haze pollution.

       OA composition was relatively similar between clean periods and polluted episodes during NHP and APEC, and the contributions of POA and SOA were also close between NHP and APEC. This is consistent with our previous conclusion that POA and SOA showed similar reductions at high altitudes during APEC (Chen et al., 2015). The improved ME-2 analysis

however showed some changes in POA and SOA composition. For example, the contributions of FFOA and OOA showed increases while COA was decreased during APEC, indicating the impacts of regional emission controls on different types of POA. OA composition during HP showed more differences between clean periods and polluted episodes. As shown in Fig. 10b, LO-OOA and BBOA showed large increases in polluted episodes for example, from 17–19% to 24–27% for LO-OOA, and from 5–8% to 9–11% for BBOA. However, such changes were not observed during NHP. Therefore, the results here

might indicate that the polluted meteorological conditions (high RH and low O$_3$) during HP facilitate the transformation of combustion-related semi-volatile species into particle phase. In comparison, OOA showed higher contributions to OA in clean periods than polluted episodes during HP. It is interesting that the FFOA and COA contributions at 260 m were comparable between clean periods and polluted episodes, which is largely different from previous observations at ground site where OA comprised much higher COA associated with a large decrease in coal combustion OA during clean periods (Sun et

al., 2013b; Sun et al., 2014).

       We further calculated the changes of aerosol species during APEC and HP compared with those during NHP. As shown in Fig. 11, almost all species decreased substantially by 40%–70% in polluted days during APEC while the reductions were less significant in clean days when air mass mainly originated from the north-northwest. This is consistent with the fact that strict emission controls were mainly implemented in the south of Beijing during APEC. However, we found an increase in

FFOA and much less reductions of SO$_2$ (~20%) and chloride during APEC compared to other species. One of the major reasons is likely due to the residential coal combustion emissions in Beijing surrounding regions (the average temperature was ~9 $^o$C) that were transported to the high altitude in urban Beijing during APEC. Consistently, the primary species related to coal combustion emissions e.g., FFOA, SO$_2$, Chl, and LO-OOA were elevated significantly from NHP to HP although the



enhancements were more dramatic in polluted episodes. For example, FFOA was increased by a factor of ~4–5 and the other three species (Chl, LO-OOA and $SO_2$) by a factor of ~2 in polluted episodes during HP. Our results suggest that coal combustion emission is the major source affecting aerosol composition at high altitudes in heating season. The remarkable enhancement of LO-OOA further supports our conclusion that LO-OOA was likely a combustion related SOA.

Comparatively, nitrate and OOA were two species showing the largest decrease during HP mainly due to the reduced photochemical production associated with lower RH and $T$.

### 3.4 Back trajectory analysis

The response of aerosol chemistry to different source regions was further demonstrated by comparing the aerosol compositions from similar air masses during the four periods. Figure 12 presents the average chemical compositions of

10 $NR-PM_1$ species and OA factors corresponding to the four clusters that were determined from the 48 h back trajectories using the Hybrid Single Particle Lagrangian Integrated Trajectory (HYSPLIT, NOAA) model (Draxler and Hess, 1997). The air mass originated from the north/northwest (C1, 40% of the time) but circulated around the south of Beijing showed the highest aerosol loading among the four clusters with the average mass concentration of $NR-PM_1$ ranging from 38.6 to 114.0 $\mu g\ m^{-3}$. Comparatively, the westerly cluster (C2) presented significantly lower aerosol loadings (28.1–62 $\mu g\ m^{-3}$), and the two

northwesterly clusters (C3 and C4) showed the lowest mass loadings (< 10 $\mu g\ m^{-3}$ for most of the time). Such large differences in aerosol loadings associated with different air masses have also been reported previously in Beijing (Han et al., 2017), which is consistent with the spatial distributions of emission sources in north China (Zhao et al., 2012). Compared with NHP, C1 from the south showed the largest reductions in $NR-PM_1$ (56%) during APEC while the reductions were much less for the other three clusters. This is consistent with the fact that strict emission controls were mainly implemented in the

regions to the south of Beijing (Chen et al., 2015). These results also highlight that emission control in the south of Beijing is the most effective measure to improve air quality in Beijing. It is interesting to note that the $NR-PM_1$ showed ubiquitously higher mass concentrations during HP1 and HP2 than NHP once the source regions were synchronized, further supporting the impacts of coal combustion emissions on PM levels. Also, the $NR-PM_1$ from the most polluted cluster (C1) showed similar levels between HP1 (114 $\mu g\ m^{-3}$) and HP2 (106 $\mu g\ m^{-3}$) while it had more differences (up to a factor of 2) for the

other three clusters indicating more source variability from relatively clean regions. The air masses dependent OA mass concentrations were similar to those of $NR-PM_1$.

$NR-PM_1$ and OA compositions vary differently among different clusters. For example, nitrate dominated SIA (28–31%) during NHP and APEC in C1 and C2, while sulfate were more prevalent in C3 and C4 (20–24%), indicating the very different emission sources between southerly and northerly air masses. In fact, the regions of C1 and C2 show significantly

high $NO_x$ emissions than C3 and C4 (Zhao et al., 2013a). Another possible explanation is the more evaporative loss of nitrate during the long-distance transport in C3 and C4, while sulfate can be transported for a longer distance because it is non-volatile. In addition, less available $NH_3$ can also be an explanation. However, such nitrate/sulfate differences among



different clusters were much reduced during HP1 and HP2 which were likely mainly due to the weaker photochemical production and less evaporative loss under lower $T$ in the heating season. OA dominated NR-PM$_1$ for all clusters, on average accounting for 46–65%. Note that the OA contributions were overall higher in C3 and C4 than C1, indicating an enhanced role of OA in PM during relatively clean periods. Despite the large differences in NR-PM$_1$ composition, we found that OA composition among different clusters were rather similar. Moreover, all clusters showed similar changes in SOA and POA compositions from NHP to HP. For example, OOA showed large decreases from 55–65% during NHP to 38–57% during HP1 and 27–40% in HP2, while the combustion-related FFOA, BBOA, and LO-OOA all showed corresponding increases during HP. Our results suggest that coal combustion emission from different source regions can all have great impacts on OA composition in heating season in Beijing.

## 4    Conclusions

We presented an analysis of submicron aerosols at a high altitude (260 m) in urban Beijing from 10 October 2014 to 18 January 2015. This study contains three periods with different emission scenarios, i.e., NHP, APEC, and HP, providing an experimental opportunity to investigate the response of aerosol chemistry to emission changes at a high altitude. The average mass concentration of NR-PM$_1$ was 64.9 μg m$^{-3}$ during NHP, and 74.5 and 48.6 μg m$^{-3}$, respectively during the two different heating periods, i.e., HP1 and HP2. Our results indicate that the average PM level in heating season is not always higher than that in non-heating season. With similar emission sources, e.g., HP1 vs. HP2, meteorological conditions can cause an average difference of NR-PM$_1$ by 35%, varying from 26 to 44% for different aerosol species. The NR-PM$_1$ composition was dominated by organics (46%) followed by nitrate (28%) and sulfate (14%) during NHP. However, the nitrate contribution in heating season was decreased substantially from 28% to ~15–18% due to reduced photochemical processing in winter while that of organics was increased to ~51–54%, and chloride was enhanced by a factor of 2 as a result of enhanced coal combustion emissions.

The sources of OA at 260 m were investigated with ME-2 using FFOA, COA, and BBOA resolved at the ground site as constrains. We observed significant changes in both POA and SOA composition from NHP to HP. Not surprising, FFOA showed the largest enhancement from 5% to 13–21% as a result of enhanced coal combustion emissions. Comparatively, OOA decreased substantially from 56% to 32–40% during HP. The less oxidized OOA however showed a remarkable increase from 9–16% to 24–26%. Our results illustrated the different properties of the two types of SOA at 260 m. While OOA was dominantly from photochemical production, LO-OOA was more likely a combustion-related SOA, consistent with the tight correlations with chloride and CO during HP. We also found a considerable contribution of COA at high altitude (10–13%) although it was nearly twice lower than that at ground site. Overall, SOA dominated OA at 260 m, on average accounting for 72% in non-heating season and 58 –64% in heating season. Compared with ground measurements, SOA showed much higher contributions at 260 m by 15–34%, demonstrating the importance of SOA in PM pollution at high altitudes in the city. Our results illustrated the large differences in POA and SOA between ground level and 260 m which



have significant implications that measurements at high altitudes are critical to better validate the simulations of POA and SOA in chemical transport models.

**Acknowledgments**

This work was supported by the National Key Project of Basic Research (2014CB447900), the Beijing Natural Science
5   Foundation (8161004), and the National Natural Science Foundation of China (41571130034, 41575120).



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



Table 1: Summary of average meteorological parameters, NR-PM$_1$ species, OA factors and gaseous species for different periods (i.e., NHP, APEC, HP1 and HP2).

|  | NHP | APEC | HP1 | HP2 |
|---|---|---|---|---|
| Meteorological parameters |  |  |  |  |
| RH | 47.1 | 29.9 | 42.7 | 29.1 |
| $T$ | 13.3 | 9.0 | 5.3 | -0.3 |
| WS | 4.2 | 4.8 | 3.7 | 5.0 |
| NR-PM$_1$ species ($\mu g\ m^{-3}$) |  |  |  |  |
| NR-PM$_1$ | 64.9 | 25.0 | 74.5 | 48.6 |
| Org | 30.1 | 12.2 | 37.9 | 26.5 |
| SO$_4$ | 8.8 | 2.5 | 11.2 | 6.9 |
| NO$_3$ | 17.9 | 7.1 | 13.3 | 7.4 |
| NH$_4$ | 5.7 | 2.2 | 5.8 | 4.3 |
| Chl | 2.5 | 1.1 | 6.3 | 3.6 |
| OA ($\mu g\ m^{-3}$) |  |  |  |  |
| FFOA | 1.4 | 1.2 | 4.7 | 5.2 |
| COA | 3.7 | 1.2 | 4.8 | 2.7 |
| BBOA | 3.0 | 0.8 | 3.2 | 2.7 |
| LO-OOA | 4.6 | 1.1 | 8.5 | 6.6 |
| OOA | 16.1 | 7.2 | 14.3 | 8.0 |
| Gaseous species |  |  |  |  |
| CO (ppm) | N/A | N/A | 3.8 | 2.5 |
| SO$_2$ (ppb) | 8.0 | 6.3 | 12.6 | 13.9 |



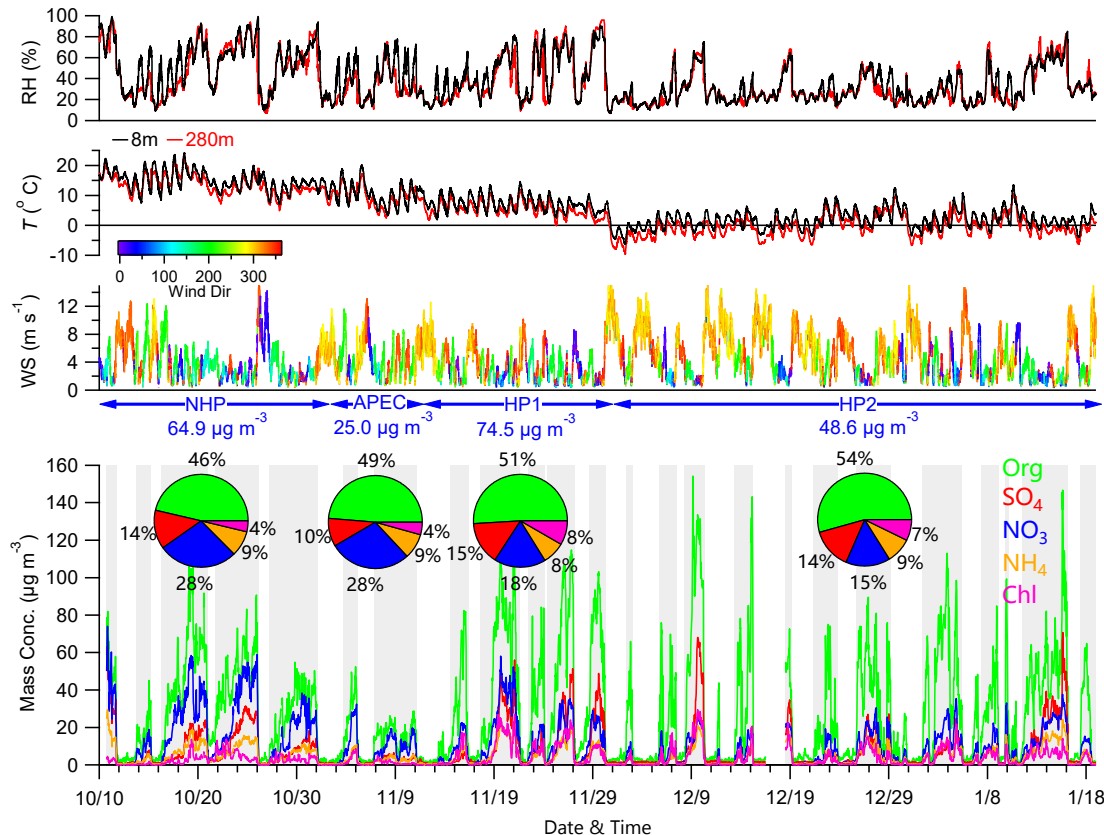

**Figure 1: Time series of meteorological parameters (RH, *T*, WS and WD) and mass concentrations of NR-PM₁ species (Org, SO₄, NO₃, NH₄, Chl). The four pie charts show the average chemical composition during the four different periods, i.e., NHP, APEC, HP1, and HP2, respectively. The polluted episodes are marked in grey for further discussions.**



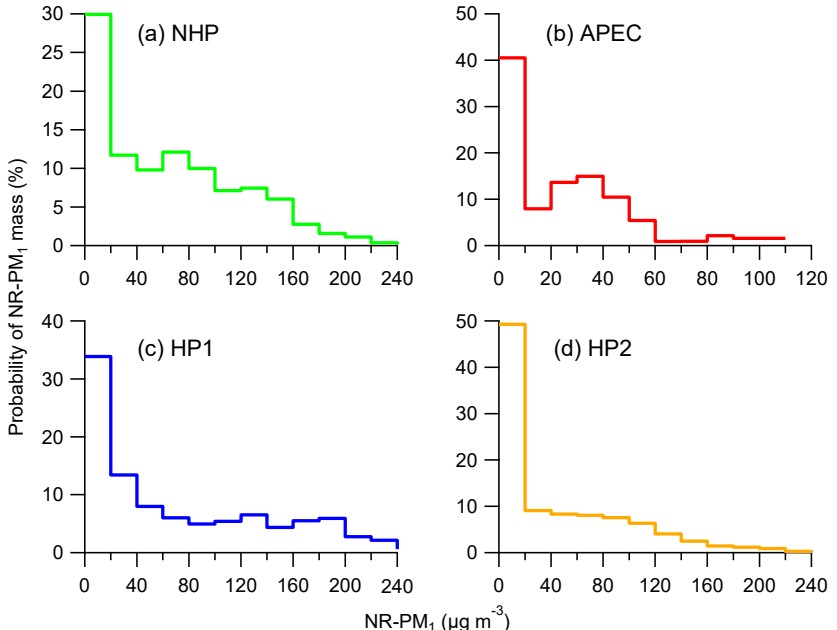

Figure 2: Probability of NR-PM$_1$ mass during four different periods (a-d), i.e., NHP, APEC, HP1, and HP2.





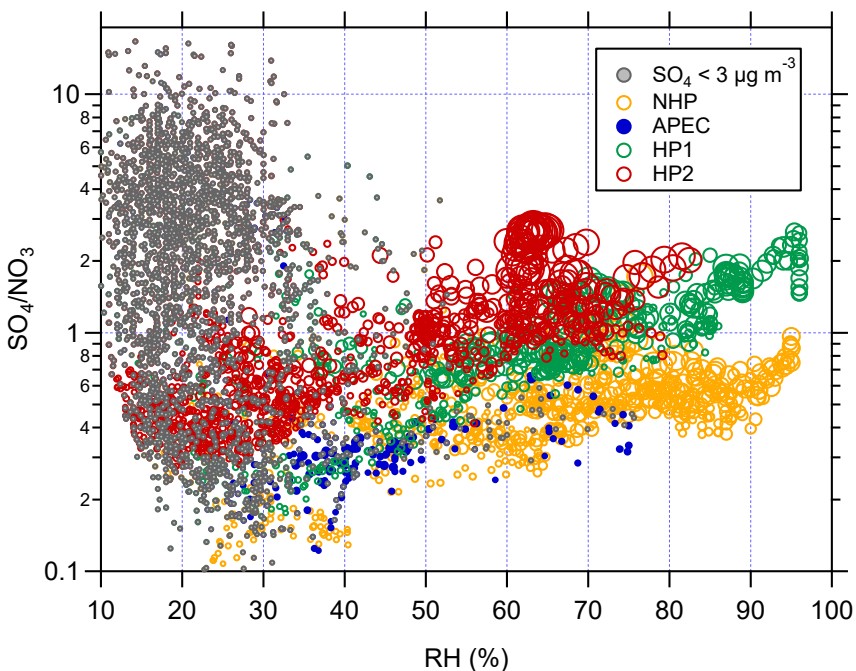

**Figure 3: The SO$_4$/NO$_3$ ratio as a function of RH during four different periods, i.e., NHP, APEC, HP1, and HP2. The points with SO$_4$ less than 3 μg m$^{-3}$ are marked in grey.**

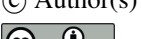



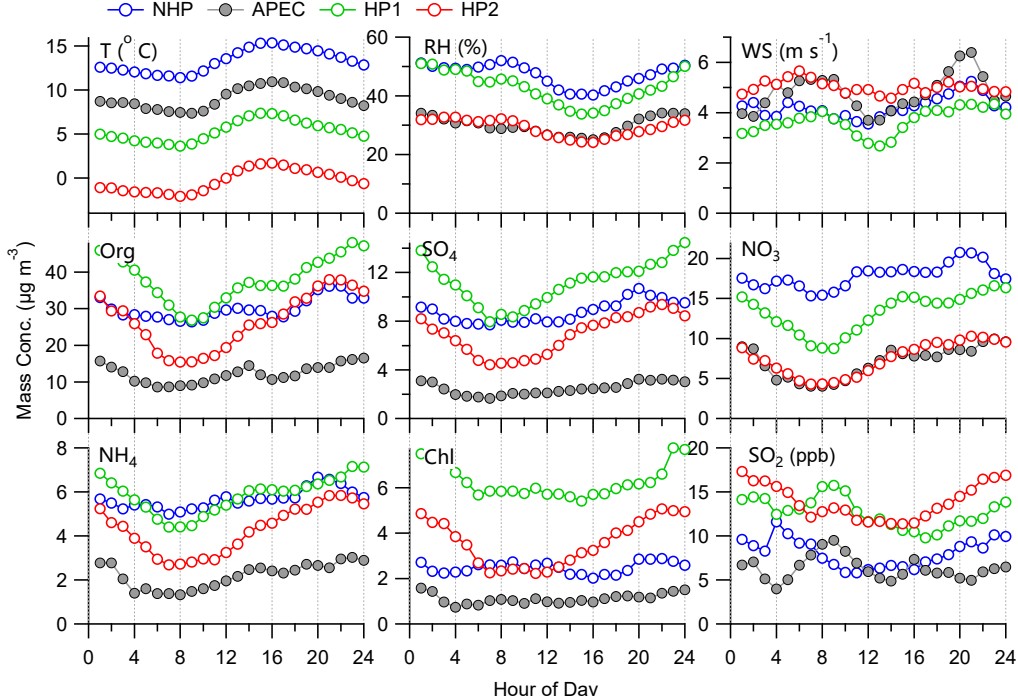

**Figure 4: Diurnal variations of meteorological parameters (RH, _T_, WS), NR-PM₁ species and SO₂ during the four different periods, i.e., NHP, APEC, HP1, and HP2.**



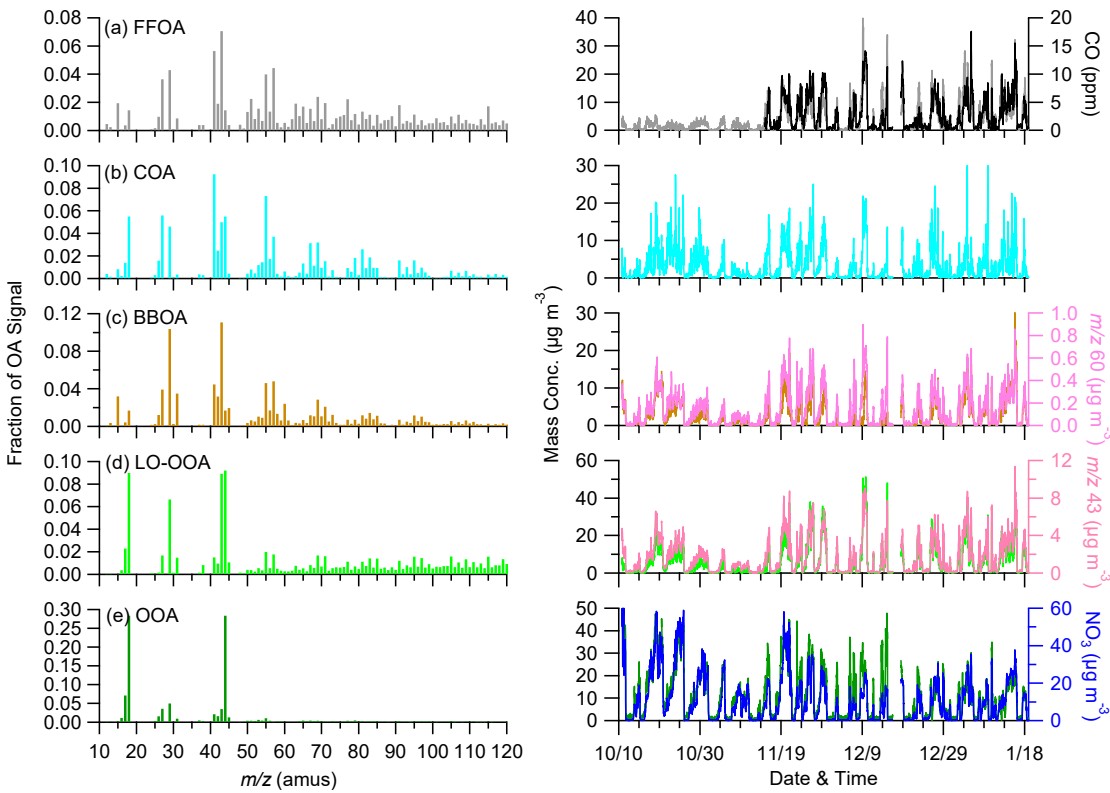

**Figure 5: Mass spectra (left panel) and time series (right panel) of five organic aerosol (OA) factors including (a) fossil fuel-related OA (FFOA), (b) cooking OA (COA), (c) biomass-burning OA (BBOA), (d) less oxidized oxygenated OA (LO-OOA), and (e) oxygenated OA (OOA). Also shown in the right panel are the time series of tracers including CO, *m/z* 60, *m/z* 43 and nitrate.**





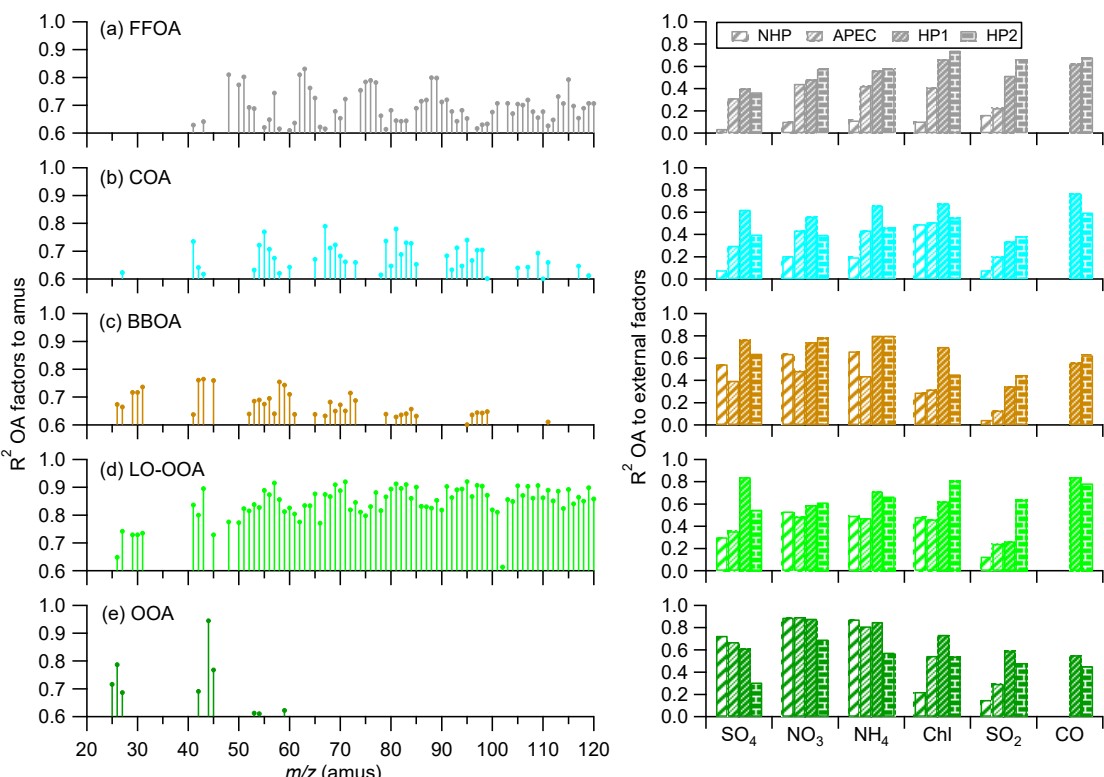

**Figure 6: The correlations between the five OA factors and *m/z*'s (left panel) and external tracer species (right panel) during the four different periods, i.e., NHP, APEC, HP1, and HP2.**





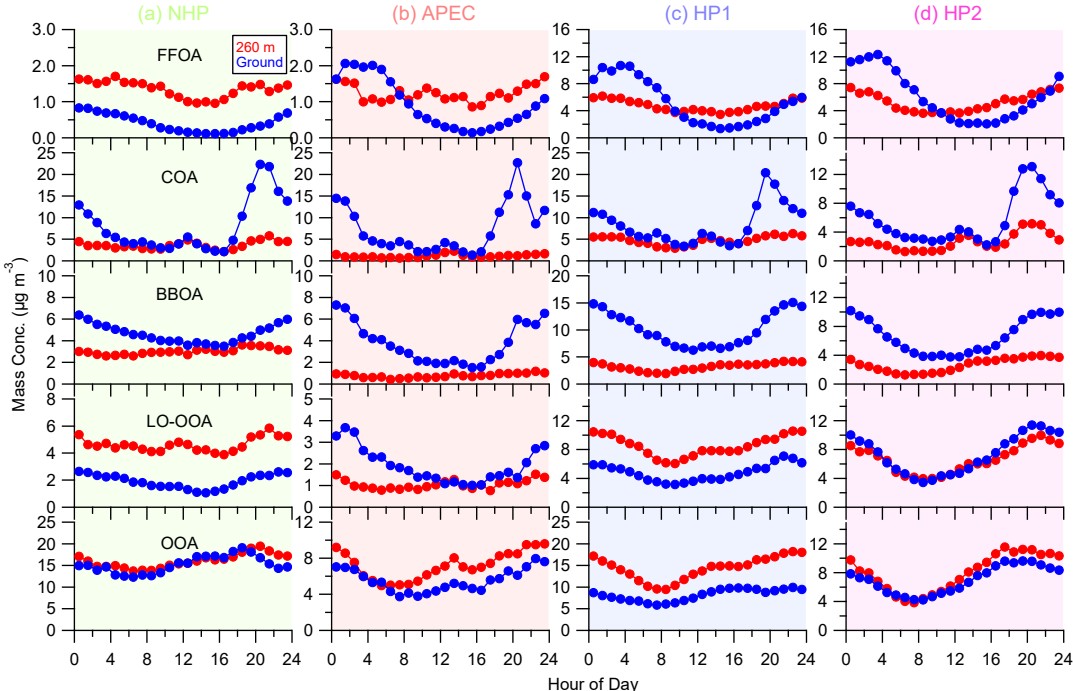

**Figure 7: Diurnal evolutions of OA factors measured during the four different periods (a-d), i.e., NHP, APEC, HP1, and HP2. The OA factors resolved at ground site are also shown for comparisons.**





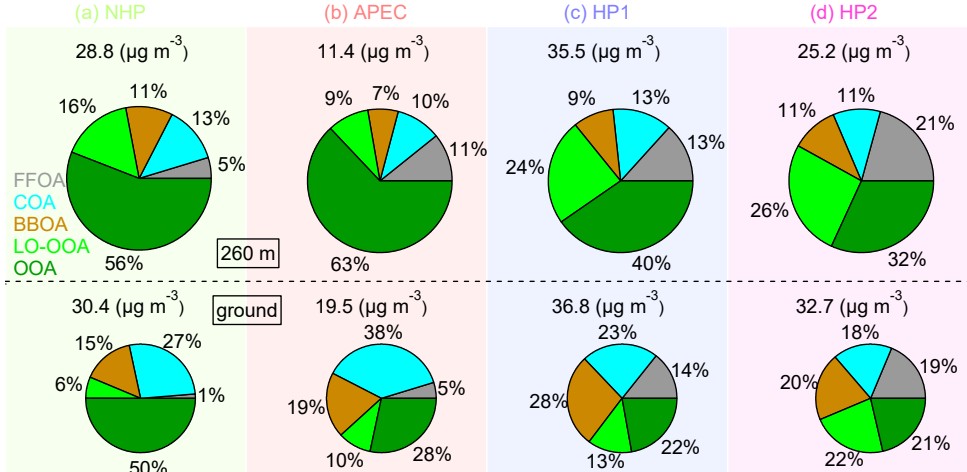

**Figure 8:** Average composition of OA during the four different periods (a-d), i.e., NHP, APEC, HP1, and HP2. The OA factors resolved at ground site are also shown for comparisons.











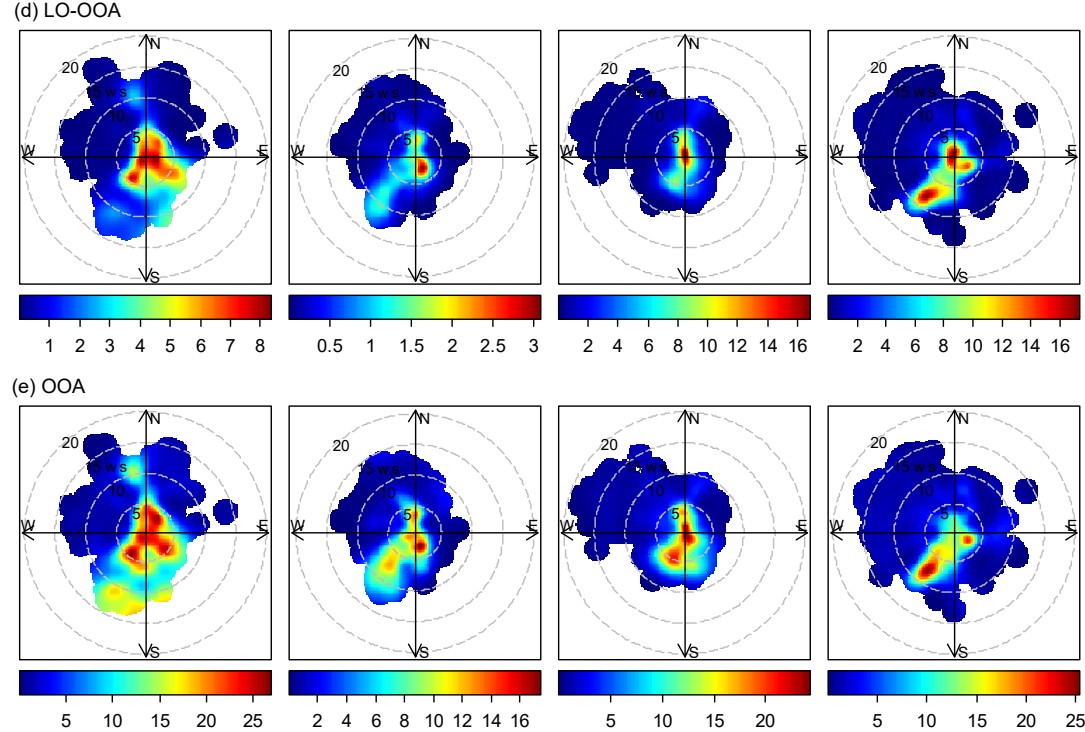

**Figure 9: Bivariate polar plots of the mass concentrations of five OA factors (a-e) at 260 m as functions of wind direction and wind speed (m s$^{-1}$) during the four different periods.**



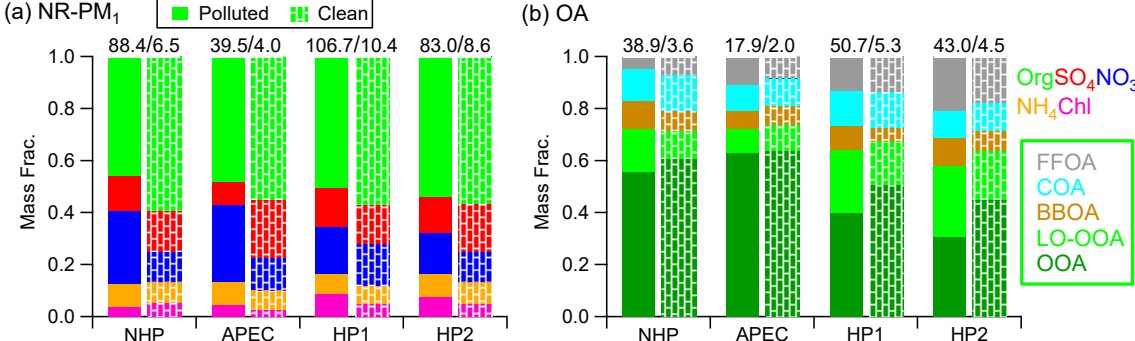

**Figure 10: Comparisons of chemical composition of (a) NR-PM₁ and (b) OA between clean periods and polluted episodes that are marked in Fig. 1 during the four different periods, i.e., NHP, APEC, HP1, and HP2. The numbers on the top of the bar graphs are the average mass concentrations of NR-PM₁ and OA.**



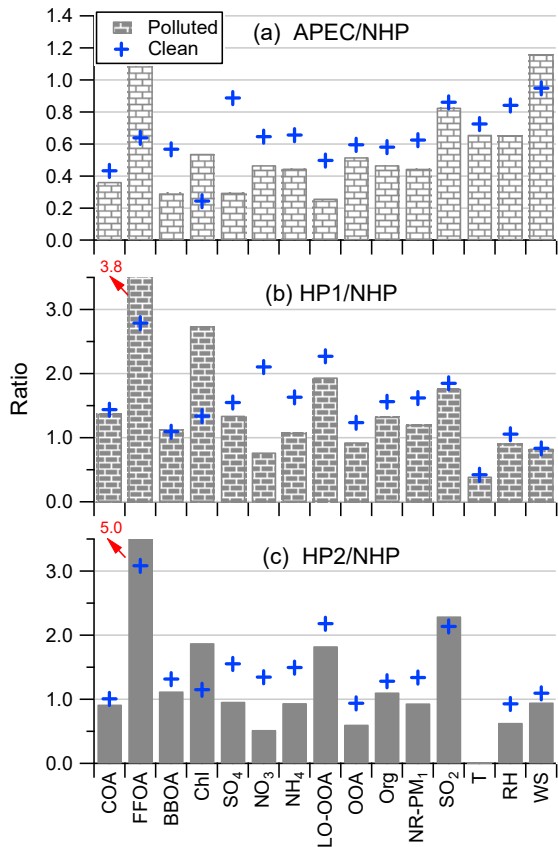

**Figure 11: Comparisons of aerosol compositions, gaseous precursors, OA factors and meteorological variables between (a) APEC and NHP, (b) HP1 and NHP and (c) HP2 and NHP. All the data were separated into clean events and pollution episodes that are marked in Fig. 1.**





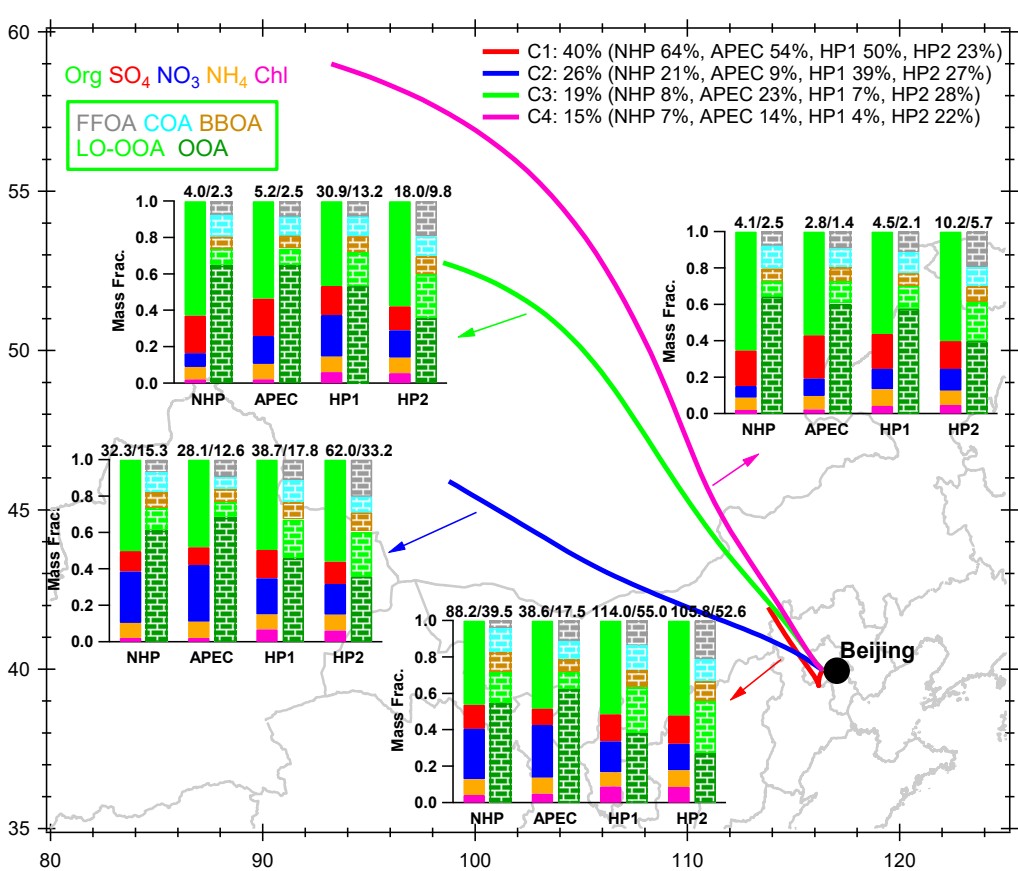

**Figure 12:** The average NR-PM$_1$ and OA composition for each cluster during the four different periods, i.e., NHP, APEC, HP1, and HP2. The numbers on the top of the bar graphs are the average mass concentrations of NR-PM$_1$ and OA. In addition, the number of trajectories and the corresponding percentages of the total trajectories are also shown in the legends.