# Peer review of "Characterization and source apportionment of organic aerosol at 260 m on a meteorological tower in Beijing, China"

_Atmospheric Chemistry and Physics, 2017_

## Referee Comment (RC1) · Anonymous Referee #3 · 11 Jan 2018

This manuscript describes atmospheric submicron aerosol sources and processes based on a field on-line measurement at an altitude of 260 nm in polluted Beijing, China, along with a comparison with an in-situ ground measurement. China has been suffering from serious air pollution issues, mainly due to complex and unclear vertical-dependent chemical and physical processes of atmospheric aerosols, although the ground-based characterization has been relatively well understood. This manuscript is well-written and provides some new and interesting data sets for understanding ambient primary and secondary organic aerosol sources and processes above an urban canopy. I strongly believe those results can make some important implications for atmospheric chemistry and physics community, and even for understanding the mechanism of haze formation in China. I recommend this paper can be published in ACP after addressing a few minor areas as follows.

1. A five-factor solution for source apportionment of organic aerosol was chosen in this study, which includes three primary factors (FFOA, COA, and BBOA), and two secondary factors (LO-OOA, and OOA). The FFOA factor here involves fossil fuel combustion sources relative to traffic and coal combustion. This should be reasonable since it could not be separated even by HR-AMS PMF approach. To further support this reasonable factor of FFOA, did the authors try to check the ratio range between FFOA and delta CO (measured total CO minus background CO) at the ground site, as comparing with any previous results (e.g., HOA+CCOA vs delta CO)? Another way to check this factor, it might be possible to constrain HOA and CCOA factors only for HR-AMS data using external reference mass spectra from previous studies at the same sampling site (e.g., Sun et al., 2016 ACP). Then, the authors could make an evaluation for unconstrained FFOA and constrained HOA+CCOA.

2. Generally, ambient oxygenated organic aerosol (OOA) derived from the AMS/ACSM PMF (or ME-2) approach includes a subset of oxidized organic aerosol factors (e.g., less or more oxidized OOA). Why did not the authors use a term of MO-OOA for your "OOA" factor, as explained in page 9 lines 25-26 "...indicating that OOA was more oxidized than LO-OOA"? (I guess it might be ok if just following the f44-f43-based criteria).

3. Page 9, lines 29-31 and page 10 lines 1-5. These are an interesting finding. The authors found that LO-OOA may be a kind of SOAs from combustion-related source(s), which has a good correlation with chloride and CO, respectively. The LO-OOA concentration at 260 m can be two times higher than that at ground site during some periods. Are these able to explain that the observation at 260 m could be closer to combustion-related SOA plumes/favorable heights rather than ground site? On the other hand, does this make sense to explain the rapid transformation of partial POA into LO-OOA (freshly formed SOA) due to processes of evaporation, oxidation and/or

re-condensation (Robinson et al., 2007 Science) during transports from ground levels?

4. Page 22, Figure 1: Some RH peaks at the ground level are much higher than at 280 m, e.g., October 14-20 and Nov. 02-09, as well as some similar peaks in heating periods, but air temperature is almost same. Is it possible to find any evidence about the enhancement of SOA productions due to aqueous-phase chemistry during these typical cases, with a comparison between ground and 260 m dataset?

5. Page 24, Figure 3: What are the data points size-scaled by?

6. Page 26, Figure 5 (left panel): The factors of FFOA, COA, and BBOA were identified using the constrain mode (a-value), but LO-OOA and OOA were resolved using the PMF free mode. So, to be more directly clear for readers, the authors may consider adding the corresponding label in each mass spectrum of POA factors (e.g., constrained or a specific a-value) and SOA factors (e.g., unconstrained or free).

7. Supplement: Pages 5-6, Figure S3 (d and e): Since LO-OOA and OOA factors were resolved by the PMF free mode, whereas FFOA, COA, and BBOA were constrained. The authors may highlight that the specific a-value is for constrained POA factors, but not for unconstrained SOA factors in both LO-OOA and OOA mass spectra.

---

## Referee Comment (RC2) · Anonymous Referee #1 · 19 Jan 2018

**Characterization and source apportionment of organic aerosol at 260 m on a meteorological tower in Beijing, China**

W. Zhou, Q. Wang, X. Zhao, W. Xu, C. Chen, W. Du, J. Zhao, F. Canonaco, A. S. H. Prévôt, P. Fu, Z. Wang, D. R. Worsnop, Y. Sun

*Atmos. Chem. Phys. Discuss., doi 10.5194/acp-2017-1039*

**Anonymous referee #1**

**General comments:**

This manuscript reports results obtained during a measurement campaign undertaken at Beijing between October 2014 and January 2015. The authors deployed an Aerodyne ACSM and a few co-located instruments ($SO_2$ and CO analyzers, meteorological data) to measure the concentration and chemical composition of NR-$PM_1$ on the top of the Beijing meteorological tower (260 m). This 3-months field campaign was divided into 3 distinct periods: the APEC summit in the middle of the campaign, during which the Chinese Government implemented strict emission control at Beijing and the surrounding regions, a non-heating period before the summit, and a heating period after the summit.

I think that the authors have a very interesting dataset in the hands, and that the measurements performed simultaneously at ground level and at 260 m height can help to better understand the evolution, dispersion, and transport of different kind of particles as a function of meteorological conditions. I would recommend the publication of this manuscript in *Atmospheric Chemistry and Physics* after the authors address the following comments.

**Specific comments:**

1) Page 4, lines 14-15: Can the authors give some clarifications about the ACSM vs. HR-ToF-AMS inter-comparison they performed during two weeks? Why did they use variable scaling factors for the correction of the ACSM data? Did they also compare PMF analysis between the two instruments? It would be also interesting if the authors mention whether the bias between their two instruments are consistent with those reported by Crenn et al. (2015) and Fröhlich et al. (2015).

2) Page 4, lines 23-24: The FFOA factor is also characterized by a high *m/z* 57/55 ratio. Moreover, given that the PMF analysis has been performed with unit mass resolution data, how do the authors know that signals at high *m/z* correspond to PAH fragments?

3) Page 8, lines 11-19: The authors mention that the diurnal pattern of FFOA was very pronounced at ground level, and much more flat at 260 m (Figure 7). Is it possible that this result is due to frequent temperature inversions which prevent vertical dispersion of FFOA, exactly like for COA (page 8, lines 28-29)?

4) Page 9, line 30 to page 10, line 1: I don't think that a high f43/f44 ratio is an argument supporting that LO-OOA corresponds to combustion-related SOA. Indeed, previous studies showed that OOA factors with high f43/f44 ratios can also correspond to factors with biogenic influences (Fig. 4 in Ng et al., 2010; Setyan et al., 2012).

**Technical comments:**

5) Page 2, line 20: "from direct emissions and secondary OA".

6) Section 2.3 (Positive matrix factorization): the authors named the two oxygenated OA factors "OOA" and "LO-OOA". Just to be consistent, I would suggest naming the first factor "MO-OOA" (more oxidized OOA) throughout the manuscript.

7) Page 5, line 2: I believe the authors wanted to say "a clear decrease of the ratio**s** of *m/z* 41/43  **and** *m/z* 55/57 as a-value increases".

8) Page 8, lines 13-14: I would suggest saying "the FFOA concentrations at ground level  **started to drop** rapidly at ~3:00–4:00".

9) Table 1: The sum of the 5 PMF factors do not match the total organic concentration reported above in the table. I'm wondering whether the authors should scale the PMF factors to the total organics.

10) Figure 1: I would suggest using a darker grey to highlight the different periods of interest.

11) Figure 4: Is it possible to include the diurnal patterns of the wind direction and CO?

12) Figure 5: I'm not sure whether it makes sense to compare the time series of BBOA and LO-OOA with *m/z* 60 and *m/z* 43, respectively. They correspond to the most representative signals of these two factors, and thus cannot be considered as external tracers. For LO-OOA, I would suggest to do the comparison with the time series of Chl, given that they presumably come from the same source (coal combustion).

13) Figure 8: Is there a reason for which the pie charts for 260 m are bigger than those for the ground site? The sizes do not seem to be related to the average concentrations.

14) Figure S3: For each time series (right panels), I would suggest to add the average concentration obtained with each a-value, as well as the correlation coefficient ($r^2$) and slope vs. the time series obtained for a-value = 0. With this information, the reader will better see how the time series deviate from the unconstrained factors when the a-value increases from 0.1 to 0.5.

**References:**

Crenn, V., Sciare, J., Croteau, P. L., Verlhac, S., Fröhlich, R., Belis, C. A., Aas, W., Äijälä, M., Alastuey, A., Artiñano, B., Baisnée, D., Bonnaire, N., Bressi, M., Canagaratna, M., Canonaco, F., Carbone, C., Cavalli, F., Coz, E., Cubison, M. J., Esser-Gietl, J. K., Green, D. C., Gros, V., Heikkinen, L., Herrmann, H., Lunder, C., Minguillón, M. C., Močnik, G., O'Dowd, C. D., Ovadnevaite, J., Petit, J. E., Petralia, E., Poulain, L., Priestman, M., Riffault, V., Ripoll, A., Sarda-Estève, R., Slowik, J. G., Setyan, A., Wiedensohler, A., Baltensperger, U., Prévôt, A. S. H., Jayne, J. T., and Favez, O.: ACTRIS ACSM intercomparison – Part 1: Reproducibility of concentration and fragment results from 13 individual Quadrupole Aerosol Chemical Speciation Monitors (Q-ACSM) and consistency with co-located instruments, Atmos. Meas. Tech., 8, 5063-5087, 10.5194/amt-8-5063-2015, 2015.

Fröhlich, R., Crenn, V., Setyan, A., Belis, C. A., Canonaco, F., Favez, O., Riffault, V., Slowik, J. G., Aas, W., Aijälä, M., Alastuey, A., Artiñano, B., Bonnaire, N., Bozzetti, C., Bressi, M., Carbone, C., Coz, E., Croteau, P. L., Cubison, M. J., Esser-Gietl, J. K., Green, D. C., Gros, V., Heikkinen, L., Herrmann, H., Jayne, J. T., Lunder, C. R., Minguillón, M. C., Močnik, G., O'Dowd, C. D., Ovadnevaite, J., Petralia, E., Poulain, L., Priestman, M., Ripoll, A., Sarda-Estève, R., Wiedensohler, A., Baltensperger, U., Sciare, J., and Prévôt, A. S. H.: ACTRIS ACSM intercomparison – Part 2: Intercomparison of ME-2 organic source apportionment results from 15 individual, co-located aerosol mass spectrometers, Atmos. Meas. Tech., 8, 2555-2576, 10.5194/amt-8-2555-2015, 2015.

Ng, N. L., Canagaratna, M. R., Zhang, Q., Jimenez, J. L., Tian, J., Ulbrich, I. M., Kroll, J. H., Docherty, K. S., Chhabra, P. S., Bahreini, R., Murphy, S. M., Seinfeld, J. H., Hildebrandt, L., Donahue, N. M., DeCarlo, P. F., Lanz, V. A., Prevot, A. S. H., Dinar, E., Rudich, Y., and Worsnop, D. R.: Organic aerosol components observed in Northern Hemispheric datasets from Aerosol Mass Spectrometry, Atmos. Chem. Phys., 10, 4625-4641, 10.5194/acp-10-4625-2010, 2010.

Setyan, A., Zhang, Q., Merkel, M., Knighton, W. B., Sun, Y., Song, C., Shilling, J. E., Onasch, T. B., Herndon, S. C., Worsnop, D. R., Fast, J. D., Zaveri, R. A., Berg, L. K., Wiedensohler, A., Flowers, B. A., Dubey, M. K., and Subramanian, R.: Characterization of submicron particles influenced by mixed biogenic and anthropogenic emissions using high-resolution aerosol mass spectrometry: results from CARES, Atmos. Chem. Phys., 12, 8131-8156, 10.5194/acp-12-8131-2012, 2012.

---

## Referee Comment (RC3) · Anonymous Referee #2 · 20 Jan 2018

The manuscript by Zhou et al. presents a detailed chemical characterization of organic aerosol at 260 m on a meteorological tower in urban Beijing by using ACSM measurements. Although the real-time measurements of aerosol particle composition at 260 m have been reported previously, this study is unique in terms of the first source apportionment analysis of OA at 260 m by using the multi-linear engine (ME-2) with the constrained POA factors identified at ground site. Fossil fuel-related OA (FFOA) dominantly from coal combustion emissions showed a large increase during heating period (HP). The SOA composition (i.e., LO-OOA and OOA) changed significantly from non-heating period (NHP) to HP. In addition, this study also observed very different OA composition between ground level and 260 m. Bivariate polar plots and back trajectory

analysis further illustrated the different source regions of OA factors in different seasons. This manuscript is generally well written and I recommend it for publication after minor revisions.

Comments: 1. 2. FFOA was still a mixture of HOA and CCOA. Did the authors try to extend the PMF solution of HR-AMS to more factors to see if HOA and CCOA can be separated? And also a comparison and discussion with previous AMS-resolved OA factors by the same group in urban Beijing during wintertime will be useful as sometimes HOA and CCOA can be separated but sometimes not. 3. Please define the polluted and clean episodes in the text. 4. Line 20-25 in Page 2, "and also highlight the importance of", change "highlight" to "highlighted". 5. Line 5-10 in Page 3, change "as a response of" to "as responses of". 6. Line 10 in Page 3, change "was all limited" to "were all limited". 7. Line 15-20 in Page 4, change "that were measured with HR-AMS" to "that was measured with HR-AMS". 8. Line 20-25 in Page 5, change "followed by a short period of clean days" to "followed by short periods of clean days" 9. Line 25 in Page 6, change "in the major mechanism" to "is the major mechanism".

---

## Author Comment (AC1) · 20 Feb 2018

We are thankful to the three reviewers for their thoughtful comments and suggestions. We have revised the manuscript accordingly. Listed below are our point-by-point responses in blue to each reviewer's comments.

**Response to Reviewer #1**

Comments:

This manuscript describes atmospheric submicron aerosol sources and processes based on a field on-line measurement at an altitude of 260 nm in polluted Beijing, China, along with a comparison with an in-situ ground measurement. China has been suffering from serious air pollution issues, mainly due to complex and unclear vertical-dependent chemical and physical processes of atmospheric aerosols, although the ground-based characterization has been relatively well understood. This manuscript is well-written and provides some new and interesting data sets for understanding ambient primary and secondary organic aerosol sources and processes above an urban canopy. I strongly believe those results can make some important implications for atmospheric chemistry and physics community, and even for understanding the mechanism of haze formation in China. I recommend this paper can be published in ACP after addressing a few minor areas as follows.

We thank the reviewer's positive comments.

1.   A five-factor solution for source apportionment of organic aerosol was chosen in this study, which includes three primary factors (FFOA, COA, and BBOA), and two secondary factors (LO-OOA, and OOA). The FFOA factor here involves fossil fuel combustion sources relative to traffic and coal combustion. This should be reasonable since it could not be separated even by HR-AMS PMF approach. To further support this reasonable factor of FFOA, did the authors try to check the ratio range between FFOA and delta CO (measured total CO minus background CO) at the ground site, as

comparing with any previous results (e.g., HOA+CCOA vs delta CO)? Another way to check this factor, it might be possible to constrain HOA and CCOA factors only for HR-AMS data using external reference mass spectra from previous studies at the same sampling site (e.g., Sun et al., 2016 ACP). Then, the authors could make an evaluation for unconstrained FFOA and constrained HOA+CCOA.

Good points. It is very challenging to separate the traffic-related HOA from coal combustion OA (CCOA) although Sun et al. (2016) was able to separate HOA from CCOA by using PMF analysis of high resolution mass spectra and the UMR spectra to $m/z = 350$. The reasons include: (1) very similar spectral patterns between HOA and CCOA at $m/z < 120$; (2) very similar temporal variations and diurnal cycles; (3) limited sensitivity of the ACSM, particularly for $m/z$'s > 50 with large uncertainties in ion transmission efficiencies. As the reviewer mentioned, the ME-2 analysis of HR-AMS by constraining HOA and CCOA factors for OA spectra might be a choice to check FFOA. However, constraining HOA and CCOA in ME-2 may introduce new ambiguities to the OA source apportionment considering the reasons above. Although they are forced to be separated, the accurate concentrations of HOA and CCOA are difficult to be evaluated. In this study, we further tried to extend the PMF solution of the HR-AMS to 6 or 7 factors, but it is difficult to obtain more interpretable and meaningful solutions, moreover, HOA and CCOA are still not separated. Without additional measurements, the 5-factor solution of HR-AMS might be a safer choice.

Following the reviewer's suggestions, we compared the ratios between FFOA and delta CO at the ground site with those reported in previous studies. FFOA/ΔCO ranges from 0.1 to 9.9 (on average 1.9) in this study, which is lower than those during November 2011 to January 2012 (1.0−27.6, on average 7.1) (Sun et al., 2013), and January 2013 (0.1−37.8, on average 8.0) (Sun et al., 2014) at the same sampling site. But it is close to that observed in Zhang et al. (2016) in December 2014 (on average 2.7). These results suggest that the FFOA/ΔCO ratios might have significant variability year by year. Another reason is that the measurements in this study were

conducted at 260 m, which might have large differences from those at ground sites.

2. Generally, ambient oxygenated organic aerosol (OOA) derived from the AMS/ACSM PMF (or ME-2) approach includes a subset of oxidized organic aerosol factors (e.g., less or more oxidized OOA). Why did not the authors use a term of MO-OOA for your "OOA" factor, as explained in page 9 lines 25-26 "...indicating that OOA was more oxidized than LO-OOA"? (I guess it might be ok if just following the f44-f43-based criteria).

Thanks the reviewer's suggestion. Such an OOA factor is typically called as MO-OOA in previous studies. However, our recent study (Sun et al., 2016a) showed that higher $f_{44}$ (fraction of $m/z$ 44 in OA) does not necessarily correspond to higher oxygen-to-carbon (O/C) ratio. Also, this OOA factor was better correlated with nitrate than sulfate in our study, which is generally different from previous findings that MO-OOA was better correlated with sulfate. Therefore, the second SOA factor was named as OOA rather than MO-OOA to avoid confusions.

3. Page 9, lines 29-31 and page 10 lines 1-5. These are an interesting finding. The authors found that LO-OOA may be a kind of SOAs from combustion-related source(s), which has a good correlation with chloride and CO, respectively. The LO-OOA concentration at 260 m can be two times higher than that at ground site during some periods. Are these able to explain that the observation at 260 m could be closer to combustion-related SOA plumes/favorable heights rather than ground site? On the other hand, does this make sense to explain the rapid transformation of partial POA into LO-OOA (freshly formed SOA) due to processes of evaporation, oxidation and/or re-condensation (Robinson et al., 2007 Science) during transports from ground levels?

Thank the reviewer's comments. Yes, our results showed that LO-OOA was more easily formed at higher heights (e.g., 260 m) during the polluted periods with higher RH and coal combustion emissions (e.g., NHP and HP1). As the reviewer pointed out,

one reason is that aerosols at 260 m are subject to more influences from regional transport, and the other reason is that the higher RH and lower $T$ at 260 m facilitate the gas-particle partitioning. However, it is very challenging to conclude that the differences of LO-OOA between 260 m and ground site are caused by evaporation, oxidation and/or re-condensation processes during the vertical transport. One major reason is the difficulties to separate the relative contributions of regional transport and vertical transport from ground site to LO-OOA at 260 m. Despite this, the reviewer suggested a very good point which should be addressed in future studies by conducting the continuously vertical measurements (from ground to 260 m) using one aerosol mass spectrometer.

4. Page 22, Figure 1: Some RH peaks at the ground level are much higher than at 280 m, e.g., October 14-20 and Nov. 02-09, as well as some similar peaks in heating periods, but air temperature is almost same. Is it possible to find any evidence about the enhancement of SOA productions due to aqueous-phase chemistry during these typical cases, with a comparison between ground and 260 m dataset?

Good points. We carefully checked the NR-PM$_1$ mass concentrations and fractions during the typical periods with much higher RH at ground level than 260 m to investigate the vertical differences and SOA formation process.

Although the nighttime RH at ground level was higher than that at 260 m during the measurement period in Fig. R1, the concentrations of SOA (LO-OOA + OOA) are consistently lower than those at 260 m. We are expecting more aqueous-phase processing at ground site due to higher RH. However, the higher SOA at 260 m suggests that there could be more important factors influencing the vertical differences. We did observe large increases in OOA and sulfate during Ep5 with the highest RH, indicating that aqueous-phase processing might be important for this episode. However, the aqueous SOA factor cannot be resolved from OOA with PMF or ME-2 analysis. This is also consistent with the fact that OOA in this study was better correlated with NO$_3$ than SO$_4$. Another reason is the average nighttime RH at

ground site was typically below 60%, and as a result, liquid water content was not high enough for strong aqueous-phase processing.

[Figure]

Figure R1. The time series of meteorological conditions (RH and $T$), NR-PM$_1$ species and OA factors both at ground and 260 m during 14 October to 20 October. Five episodes with much higher RH at ground level than 260 m are marked in dark grey.

[Figure]

Figure R2. Average RH, mass concentrations and fractions for NR-PM$_1$ species and OA factors at ground level and 260 m during the five episodes marked in Fig. R1.

[Figure]

Figure 3: The variations of SO$_4$/NO$_3$ ratios as a function of RH during four different periods, i.e., NHP, APEC, HP1, and HP2. The marker sizes indicate the SO$_4$ concentrations, and the data points with the SO$_4$ concentrations less than 3 µg m$^{-3}$ are marked in grey.

5. Page 24, Figure 3: What are the data points size-scaled by?

Thanks the reviewer for pointing this out. The data points were sized-scaled by the mass concentrations of sulfate ($SO_4$). The data points with $SO_4$ less than 3 µg m$^{-3}$ are marked in grey. We revised this figure in the new version of the manuscript as shown above.

6. Page 26, Figure 5 (left panel): The factors of FFOA, COA, and BBOA were identified using the constrain mode (a-value), but LO-OOA and OOA were resolved using the PMF free mode. So, to be more directly clear for readers, the authors may consider adding the corresponding label in each mass spectrum of POA factors (e.g., constrained or a specific a-value) and SOA factors (e.g., unconstrained or free).

Good points. We revised Figure 5 as below.

[Figure]

Figure 5: Mass spectra (left panel) and time series (right panel) of five organic aerosol (OA) factors including (a) fossil fuel-related OA (FFOA), (b) cooking OA (COA), (c) biomass-burning OA (BBOA), (d) less oxidized oxygenated OA (LO-OOA), and (e) oxygenated OA (OOA). Also shown in the right panel are the time series of tracers including CO, nitrate, $m/z$ 60, and $m/z$ 43.

7. Supplement: Pages 5-6, Figure S3 (d and e): Since LO-OOA and OOA factors were resolved by the PMF free mode, whereas FFOA, COA, and BBOA were constrained. The authors may highlight that the specific a-value is for constrained POA factors, but not for unconstrained SOA factors in both LO-OOA and OOA mass spectra.

We thank the reviewer for these suggestions. We revised the caption of Figure S3 to make the figure more clear for the readers. Now it reads:

"Figure S3: Mass spectra (left panel) and time series (right panel) of five organic aerosol (OA) components resolved at 260 m by ACSM using multi-linear engine 2 (ME-2): (a) fossil fuel-related OA (FFOA), (b) cooking OA (COA), (c) biomass-burning OA (BBOA), (d) low-oxidized oxygenated OA (LO-OOA), and (e) oxygenated OA (OOA). The 4-factor solution of PMF result was also shown here. Note that the mass spectra of two SOA factors in (d) and (e) were unconstrained, and the a values refer to those of three POA factors (i.e., FFOA, COA and BBOA)."

**Response to Reviewer #2**

General comments:

This manuscript reports results obtained during a measurement campaign undertaken at Beijing between October 2014 and January 2015. The authors deployed an Aerodyne ACSM and a few co-located instruments (SO2 and CO analyzers, meteorological data) to measure the concentration and chemical composition of NR-PM1 on the top of the Beijing meteorological tower (260 m). This 3-months field campaign was divided into 3 distinct periods: the APEC summit in the middle of the campaign, during which the Chinese Government implemented strict emission control at Beijing and the surrounding regions, a non-heating period before the summit, and a heating period after the summit.

I think that the authors have a very interesting dataset in the hands, and that the measurements performed simultaneously at ground level and at 260 m height can help to better understand the evolution, dispersion, and transport of different kind of particles as a function of meteorological conditions. I would recommend the publication of this manuscript in *Atmospheric Chemistry and Physics* after the authors address the following comments.

We thank the reviewer's positive comments.

Specific comments:

1) Page 4, lines 14-15: Can the authors give some clarifications about the ACSM vs. HR-ToF-AMS inter-comparison they performed during two weeks? Why did they use variable scaling factors for the correction of the ACSM data? Did they also compare PMF analysis between the two instruments? It would be also interesting if the authors mention whether the bias between their two instruments are consistent with those reported by Crenn et al. (2015) and Fröhlich et al. (2015).

Good point. We did a two-week inter-comparison between HR-ToF-AMS and ACSM measurements before this study. The ACSM measurements were further corrected using the regression slopes against HR-ToF-AMS measurements (0.61–1.24) from the inter-comparisons to reduce the uncertainties in vertical comparisons. As shown in Figure R3, all NR-PM$_1$ species are well correlated between the two instruments ($R^2$=0.97–1) and the slopes ranged from 0.61–1.24. The different scaling factors for different species mainly due to that the ACSM measurements can have uncertainties of 9–36% for different NR-PM$_1$ species, consistent with the results reported previously (Crenn et al., 2015). Such information was now added in section 2.2.

[Figure]

Figure R3. Inter-comparisons between ACSM and HR-ToF-AMS measurements for different NR-PM$_1$ species.

We also compared the OA factors resolved from PMF analysis of the two datasets during the inter-comparison period. As shown in Figure R4, the mass spectra and time series of three OA factors are overall similar between ACSM and HR-ToF-AMS. However, we also noticed some differences in both mass spectra and time series. In particular, HR-ToF-AMS appears to report higher COA concentrations than ACSM, while the CCOA concentrations are relatively low. Comparatively, the OOA concentrations agree well between the two instruments. Therefore, we used ME-2

analysis to constrain POA factors at 260 m using those resolved at ground site for better comparisons at the two different heights. In addition, the mass spectrum of ACSM OOA presented higher $f_{44}$ than that of HR-ToF-AMS, which is consistent with that reported by Fröhlich et al. (2015), and also explained the high $f_{44}$ in OOA spectrum in this study.

[Figure]

Figure R4. Inter-comparisons of mass spectra and time series of three OA factors from PMF analysis of ACSM and HR-ToF-AMS organic mass spectra. The red and black lines represent ACSM and HR-ToF-AMS solutions, respectively.

2) Page 4, lines 23-24: The FFOA factor is also characterized by a high $m/z$ 57/55 ratio. Moreover, given that the PMF analysis has been performed with unit mass resolution data, how do the authors know that signals at high $m/z$ correspond to PAH fragments?

Good points. PMF analysis was performed to the unit mass resolution spectra of OA ($m/z$ 12−350) that were measured with HR-ToF-AMS, which supplied the information at large $m/z$'s (>150). Figure S1 showed that resolved FFOA spectrum showed strong

PAH signatures, which are *m/z*'s 152, 165, 178, 189, 202, 215, 226, 239… (Dzepina et al., 2007), consistent with the results at the same site (Sun et al., 2016b). Hu et al. (2016) also reported the pronounced peaks of PAHs at *m/z*'s 152, 165, 178 and 189 in the CCOA spectrum in Beijing. Moreover, Sun et al. (2016b) found that CCOA was tightly correlated with PAHs, which was also observed for FFOA in this study (Figure R5). Therefore, we concluded that signals at high *m/z* correspond to PAHs fragments.

[Figure]

Figure R5. Correlations of five OA factors resolved from the HR-ToF-AMS with each unit *m/z*.

3) Page 8, lines 11-19: The authors mention that the diurnal pattern of FFOA was very pronounced at ground level, and much more flat at 260 m (Figure 7). Is it possible that this result is due to frequent temperature inversions which prevent vertical dispersion of FFOA, exactly like for COA (page 8, lines 28-29)?

Yes, we agree with the reviewer that the more flat FFOA diurnal pattern at 260 m than the ground level could be due to the frequent temperature inversions that

suppressed the vertical mixing of FFOA from local emissions. Following the reviewer's suggestions, we added such an explanation in the revised manuscript, and now it reads:

"One explanation is the frequent temperature inversions at nighttime that suppressed the vertical convection of local FFOA to high heights."

4) Page 9, line 30 to page 10, line 1: I don't think that a high f43/f44 ratio is an argument supporting that LO-OOA corresponds to combustion-related SOA. Indeed, previous studies showed that OOA factors with high f43/f44 ratios can also correspond to factors with biogenic influences (Fig. 4 in Ng et al., 2010; Setyan et al., 2012).

Thanks the reviewer's for pointing this out. We concluded that LO-OOA was a combustion-related SOA mainly because that biogenic emissions are not expected to be important in winter in Beijing.

Technical comments:

5) Page 2, line 20: "from direction emissions and secondary OA".

Thanks for the reviewer's carefulness. We revised the sentence in the revised manuscript. It now reads:

"…primary OA (POA) from direct emissions and secondary OA (SOA) from …"

6) Section 2.3 (Positive matrix factorization): the authors named the two oxygenated OA factors "OOA" and "LO-OOA". Just to be consistent, I would suggest naming the first factor "MO-OOA" (more oxidized OOA) throughout the manuscript.

Thanks the reviewer's suggestion. Such an OOA factor is typically called as MO-OOA in previous studies. However, our recent study (Sun et al., 2016a) showed that higher $f_{44}$ (fraction of $m/z$ 44 in OA) does not necessarily correspond to higher

oxygen-to-carbon (O/C) ratio. Also, this OOA factor was better correlated with nitrate than sulfate in our study, which is generally different from previous findings that MO-OOA was better correlated with sulfate. Therefore, the second SOA factor was named as OOA rather than MO-OOA to avoid confusions.

7) Page 5, line 2: I believe the authors wanted to say "a clear decrease of the ratio**s** of *m/z* 41/43  **and** *m/z* 55/57 as a-value increases".

Yes. We have revised the sentence following the reviewer's suggestions. It now reads:

"…the FFOA spectrum showed a clear decrease of the ratios of *m/z* 41/43 and *m/z* 55/57 as a-value increases …"

8) Page 8, lines 13-14: I would suggest saying "the FFOA concentrations at ground level  **started to drop** rapidly at ~3:00–4:00".

Thanks for the reviewer's suggestion. We revised the sentence in the new version of the manuscript. It now reads:

"…the FFOA concentrations at ground level started to drop rapidly at ~3:00–4:00"

9) Table 1: The sum of the 5 PMF factors do not match the total organic concentration reported above in the Table. I'm wondering whether the authors should scale the PMF factors to the total organics.

We thank the reviewer's comments. The differences were due to the residuals in ME-2 analysis that cannot be explained by the five OA factors. The unexplained residuals on average account for 4 – 7% of total OA. PMF analysis of HR-ToF-ACSM organic spectra also have similar residuals. To be consistent, we did not scale the sum of OA factors to the total organics in this study.

10) Figure 1: I would suggest using a darker grey to highlight the different periods of interest.

Following the reviewer's suggestions, we changed the color from light grey to dark grey in Figure 1 in the revised manuscript.

11) Figure 4: Is it possible to include the diurnal patterns of the wind direction and CO?

Thanks the reviewer's suggestions. We calculated the diurnal patterns of wind direction and CO and added this figure in supplementary in the revised manuscript.

[Figure]

Figure R6. Diurnal variations of (a) wind direction and (b) CO during the four different periods, i.e., NHP, APEC, HP1, HP2. Note that the CO data were not available during NHP and APEC.

12) Figure 5: I'm not sure whether it makes sense to compare the time series of BBOA and LO-OOA with $m/z$ 60 and $m/z$ 43, respectively. They correspond to the most representative signals of these two factors, and thus cannot be considered as external tracers. For LO-OOA, I would suggest to do the comparison with the time series of Chl, given that they presumably come from the same source (coal combustion).

Thanks for the suggestions. $m/z$ 60 and $m/z$ 43 are the most representative signals of BBOA and LO-OOA, respectively, rather than the external tracers. We compared the time series of BBOA and LO-OOA with $m/z$ 60 and $m/z$ 43, respectively, to illustrate the typical mass spectral peaks of these two factors.

We agree with the reviewer that LO-OOA can be correlated with chloride that is mainly from coal combustion in winter in Beijing. Indeed, LO-OOA was well correlated with Chl during HP1 and HP2 ($R^2$=0.71-0.81). Because this study also covers non-heating season when biomass burning could be a more important source of chloride, we did not show chloride as an external tracer in Figure 5. In fact, the correlations between LO-OOA and Chl were much weaker during NHP and APEC (Figure 6). Still, the comparisons of correlations between LO-OOA and Chl were performed and shown in Figure 6 (right panel).

13) Figure 8: Is there a reason for which the pie charts for 260 m are bigger than those for the ground site? The sizes do not seem to be related to the average concentrations.

Thanks the reviewer's carefulness. Yes, the sizes are not related to the average concentrations which are already shown on the top of pie charts. Because this study is focused on OA characterization at 260 m, we make the pie charts at 260 m bigger than those at ground level to highlight the results.

14) Figure S3: For each time series (right panels), I would suggest to add the average concentration obtained with each a-value, as well as the correlation coefficient (r2) and slope vs. the time series obtained for a-value = 0. With this information, the reader will better see how the time series deviate from the unconstrained factors when the a-value increases from 0.1 to 0.5.

Good points. The average concentrations of five OA factors obtained with each a-value have been presented in Figure S2(c). Following the reviewer's suggestions, we added more visual details on the comparisons between different a-values in the revised manuscript (Table S1). A small point to note is that a-value = 0 means the complete constrained condition rather than the unconstrained mode.

**Table S1.** The average concentrations for OA factors obtained with each a-value

(Avg). Also shown are the correlation coefficients ($R^2$) and regression slopes (Slope) when a-value ranges from 0.1-0.5 versus the time series obtained for a-value = 0 for each OA factor.

|  | a-value=0 | a-value=0.1 | a-value=0.2 | a-value=0.3 | a-value=0.4 | a-value=0.5 |
|---|---|---|---|---|---|---|
| **FFOA** | Avg=3.8 | Avg=4.0 | Avg=3.6 | Avg=3.7 | Avg=3.7 | Avg=4.3 |
|  |  | $R^2$=0.99 | $R^2$=0.97 | $R^2$=0.96 | $R^2$=0.97 | $R^2$=0.96 |
|  |  | Slope=1.04 | Slope=0.95 | Slope=0.95 | Slope=0.99 | Slope=1.15 |
| **COA** | Avg=3.1 | Avg=3.3 | Avg=3.2 | Avg=3.1 | Avg=2.9 | Avg=2.9 |
|  |  | $R^2$=1.00 | $R^2$=0.96 | $R^2$=0.90 | $R^2$=0.89 | $R^2$=0.89 |
|  |  | Slope=1.04 | Slope=0.93 | Slope=0.81 | Slope=0.76 | Slope=0.77 |
| **BBOA** | Avg=2.7 | Avg=3.2 | Avg=3.7 | Avg=4.1 | Avg=4.5 | Avg=4.6 |
|  |  | $R^2$=0.99 | $R^2$=0.97 | $R^2$=0.94 | $R^2$=0.93 | $R^2$=0.91 |
|  |  | Slope=1.17 | Slope=1.36 | Slope=1.48 | Slope=1.56 | Slope=1.54 |
| **LO-OOA** | Avg=5.9 | Avg=5.5 | Avg=5.3 | Avg=5.2 | Avg=5.2 | Avg=4.9 |
|  |  | $R^2$=1.00 | $R^2$=0.97 | $R^2$=0.95 | $R^2$=0.96 | $R^2$=0.94 |
|  |  | Slope=0.91 | Slope=0.89 | Slope=0.88 | Slope=0.86 | Slope=0.78 |
| **OOA** | Avg=11.0 | Avg=10.7 | Avg=10.8 | Avg=10.5 | Avg=10.2 | Avg=9.8 |
|  |  | $R^2$=1.00 | $R^2$=1.00 | $R^2$=1.00 | $R^2$=1.00 | $R^2$=1.00 |
|  |  | Slope=0.97 | Slope=0.99 | Slope=0.96 | Slope=0.94 | Slope=0.89 |

**Response to Reviewer #3**

Comments:

The manuscript by Zhou et al. presents a detailed chemical characterization of organic aerosol at 260 m on a meteorological tower in urban Beijing by using ACSM measurements. Although the real-time measurements of aerosol particle composition at 260 m have been reported previously, this study is unique in terms of the first source apportionment analysis of OA at 260 m by using the multi-linear engine (ME-2) with the constrained POA factors identified at ground site. Fossil fuel-related OA (FFOA) dominantly from coal combustion emissions showed a large increase during heating period (HP). The SOA composition (i.e., LO-OOA and OOA) changed significantly from non-heating period (NHP) to HP. In addition, this study also observed very different OA composition between ground level and 260 m. Bivariate polar plots and back trajectory analysis further illustrated the different source regions of OA factors in different seasons. This manuscript is generally well written and I recommend it for publication after minor revisions.

We thank the reviewer's positive comments.

1.2. FFOA was still a mixture of HOA and CCOA. Did the authors try to extend the PMF solution of HR-AMS to more factors to see if HOA and CCOA can be separated? And also a comparison and discussion with previous AMS-resolved OA factors by the same group in urban Beijing during wintertime will be useful as sometimes HOA and CCOA can be separated but sometimes not.

Yes, it is very challenging to separate the traffic-related HOA from coal combustion OA (CCOA) through PMF analysis of unit mass resolution mass spectra of either HR-ToF-AMS or ACSM. The reasons include: (1) very similar spectral patterns between HOA and CCOA at $m/z$ < 120; (2) very similar temporal variations and diurnal cycles; (3) limited sensitivity of the ACSM, particularly for $m/z$'s > 50 with large uncertainties in ion transmission efficiencies. Sun et al. (2016b) was able to

separate HOA from CCOA by using PMF analysis of high resolution mass spectra and the unit mass resolution mass spectra to $m/z$ = 350, while they were not separated in this study even extending the solution to 6 or 7 factors. Therefore, the two sources are combined into one factor, i.e., fossil fuel related OA (FFOA). The FFOA spectrum pattern identified in this study resembles much more to that of smoky coal (Lin et al., 2017) than the standard traffic-related HOA (Ng et al., 2011). Consistent with our previous results in the winter of 2013–2014 (Sun et al., 2016b), FFOA showed dramatic increase after the heating season start on 15 November, and we also found strong PAH signatures in the FFOA spectrum. These results together indicated that FFOA was dominantly from the coal combustion.

3. Please define the polluted and clean episodes in the text.

Good point. The polluted episodes including the formation, evolution and cleaning stages are marked in grey in Figure 1, and the average mass concentrations of NR-PM$_1$ during polluted episodes are generally larger than 50 μg m$^{-3}$. The rest periods are defined as clean periods. It is now clarified in the revised manuscript as:

"the changes in NR-PM$_1$ were characterized by routine cycles of polluted episodes (marked in grey in Fig. 1) and clean periods (the rest of the time) during HP2"

4. Line 20-25 in Page 2, "and also highlight the importance of", change "highlight" to "highlighted".

Thanks for the reviewer's carefulness. We revised the grammatical mistake in the revised manuscript. Now it reads:

"… and also highlighted the importance of OA in the rapid formation of severe haze"

5. Line 5-10 in Page 3, change "as a response of" to "as responses of".

Thanks for pointing this out. Yes, we revised the sentence in the new version of the

manuscript. Now it reads:

"The results showed similar reductions … as responses of emission controls"

6. Line 10 in Page 3, change "was all limited" to "were all limited".

Thanks for the reviewer's carefulness. We revised the singular and plural forms in the manuscript. Now it reads:

"…PMF analyses of OA at 260 m in previous studies were all limited to…"

7. Line 15-20 in Page 4, change "that were measured with HR-AMS" to "that was measured with HR-AMS".

Thanks for the suggestions. Yes, we checked the sentence and found that this is a mistake, which lies in the "as that of ACSM" rather than "that were measured with HR-AMS". We revised it and now it reads:

"…performed to the unit mass resolution spectra of OA ($m/z$ 12−350) at ground site that were measured with HR-AMS during the same period as those of ACSM"

8. Line 20-25 in Page 5, change "followed by a short period of clean days" to "followed by short periods of clean days".

We thank the review for pointing this mistake. We revised this and now it reads:

"…followed by short periods of clean days…"

9. Line 25 in Page 6, change "in the major mechanism" to "is the major mechanism".

Yes, we revised the sentence following the reviewer's suggestion. Now it reads:

"… $SO_2$ by $NO_2$ in aerosol water is the major mechanism"

**References:**

Crenn, V., Sciare, J., Croteau, P. L., Verlhac, S., Fröhlich, R., Belis, C. A., Aas, W., Äijälä, M., Alastuey, A., Artiñano, B., Baisnée, D., Bonnaire, N., Bressi, M., Canagaratna, M., Canonaco, F., Carbone, C., Cavalli, F., Coz, E., Cubison, M. J., Esser-Gietl, J. K., Green, D. C., Gros, V., Heikkinen, L., Herrmann, H., Lunder, C., Minguillón, M. C., Močnik, G., O'Dowd, C. D., Ovadnevaite, J., Petit, J. E., Petralia, E., Poulain, L., Priestman, M., Riffault, V., Ripoll, A., Sarda-Estève, R., Slowik, J. G., Setyan, A., Wiedensohler, A., Baltensperger, U., Prévôt, A. S. H., Jayne, J. T., and Favez, O.: ACTRIS ACSM intercomparison – Part 1: Reproducibility of concentration and fragment results from 13 individual Quadrupole Aerosol Chemical Speciation Monitors (Q-ACSM) and consistency with co-located instruments, Atmos. Meas. Tech., 8, 5063-5087, 10.5194/amt-8-5063-2015, 2015.

Dzepina, K., Arey, J., Marr, L. C., Worsnop, D. R., Salcedo, D., Zhang, Q., Onasch, T. B., Molina, L. T., Molina, M. J., and Jimenez, J. L.: Detection of particle-phase polycyclic aromatic hydrocarbons in Mexico City using an aerosol mass spectrometer, Int. J. Mass Spectrom., 263, 152-170, 10.1016/j.ijms.2007.01.010, 2007.

Fröhlich, R., Crenn, V., Setyan, A., Belis, C. A., Canonaco, F., Favez, O., Riffault, V., Slowik, J. G., Aas, W., Aijälä, M., Alastuey, A., Artiñano, B., Bonnaire, N., Bozzetti, C., Bressi, M., Carbone, C., Coz, E., Croteau, P. L., Cubison, M. J., Esser-Gietl, J. K., Green, D. C., Gros, V., Heikkinen, L., Herrmann, H., Jayne, J. T., Lunder, C. R., Minguillón, M. C., Močnik, G., O'Dowd, C. D., Ovadnevaite, J., Petralia, E., Poulain, L., Priestman, M., Ripoll, A., Sarda-Estève, R., Wiedensohler, A., Baltensperger, U., Sciare, J., and Prévôt, A. S. H.: ACTRIS ACSM intercomparison – Part 2: Intercomparison of ME-2 organic source apportionment results from 15 individual, co-located aerosol mass spectrometers, Atmos. Meas. Tech., 8, 2555-2576, 10.5194/amt-8-2555-2015, 2015.

Hu, W., Hu, M., Hu, W., Jimenez, J. L., Yuan, B., Chen, W., Wang, M., Wu, Y., Chen, C., Wang, Z., Peng, J., Zeng, L., and Shao, M.: Chemical composition, sources, and aging process of submicron aerosols in Beijing: Contrast between summer and winter, J. Geophys. Res. - Atmos., 121, 1955-1977, 10.1002/2015jd024020, 2016.

Lin, C., Ceburnis, D., Hellebust, S., Buckley, P., Wenger, J., Canonaco, F., Prévôt, A. S. H., Huang, R.-J., O'Dowd, C., and Ovadnevaite, J.: Characterization of Primary Organic Aerosol from Domestic Wood, Peat, and Coal Burning in Ireland, Environ. Sci. Technol., 51, 10624-10632, 10.1021/acs.est.7b01926, 2017.

Ng, N. L., Canagaratna, M. R., Jimenez, J. L., Zhang, Q., Ulbrich, I. M., and Worsnop, D. R.: Real-Time Methods for Estimating Organic Component Mass Concentrations from Aerosol Mass Spectrometer Data, Environ. Sci. Technol., 45, 910-916, 10.1021/es102951k, 2011.

Sun, Y., Jiang, Q., Wang, Z., Fu, P., Li, J., Yang, T., and Yin, Y.: Investigation of the sources and evolution processes of severe haze pollution in Beijing in January

2013, Journal of Geophysical Research: Atmospheres, 119, 4380-4398, 10.1002/2014JD021641, 2014.

Sun, Y., Du, W., Fu, P., Wang, Q., Li, J., Ge, X., Zhang, Q., Zhu, C., Ren, L., Xu, W., Zhao, J., Han, T., Worsnop, D., and Wang, Z.: Primary and secondary aerosols in Beijing in winter: sources, variations and processes, Atmos. Chem. Phys., 16, 8309-8329, 10.5194/acp-16-8309-2016, 2016a.

Sun, Y., Du, W., Fu, P., Wang, Q., Li, J., Ge, X., Zhang, Q., Zhu, C., Ren, L., Xu, W., Zhao, J., Han, T., Worsnop, D. R., and Wang, Z.: Primary and secondary aerosols in Beijing in winter: sources, variations and processes, Atmos. Chem. Phys., 16, 8309-8329, 10.5194/acp-16-8309-2016, 2016b.

Sun, Y. L., Wang, Z. F., Fu, P. Q., Yang, T., Jiang, Q., Dong, H. B., Li, J., and Jia, J. J.: Aerosol composition, sources and processes during wintertime in Beijing, China, Atmos. Chem. Phys., 13, 4577-4592, 10.5194/acp-13-4577-2013, 2013.

Zhang, J. K., Cheng, M. T., Ji, D. S., Liu, Z. R., Hu, B., Sun, Y., and Wang, Y. S.: Characterization of submicron particles during biomass burning and coal combustion periods in Beijing, China, Sci. Total. Environ., 562, 812-821, 10.1016/j.scitotenv.2016.04.015, 2016.